# A conformational switch controlling the toxicity of the prion protein

Karl Frontzek [1,9], Marco Bardelli[2,3,9], Assunta Senatore[1,9], Anna Henzi[1], Regina R. Reimann[1], Seden Bedir [1], Marika Marino [4], Rohanah Hussain [5], Simon Jurt[6], Georg Meisl [7], Mattia Pedotti[2], Federica Mazzola[2], Giuliano Siligardi[5], Oliver Zerbe[6], Marco Losa [1], Tuomas Knowles [7], Asvin Lakkaraju[1], Caihong Zhu[1], Petra Schwarz[1], Simone Hornemann [1], Matthew G. Holt [4,8], Luca Simonelli[2], Luca Varani [2,10] ✉ and Adriano Aguzzi [1,10] ✉

Prion infections cause conformational changes of the cellular prion protein (PrP^C) and lead to progressive neurological impairment. Here we show that toxic, prion-mimetic ligands induce an intramolecular R208-H140 hydrogen bond ('H-latch'), altering the flexibility of the α2–α3 and β2–α2 loops of PrP^C. Expression of a PrP^2Cys mutant mimicking the H-latch was constitutively toxic, whereas a PrP^R207A mutant unable to form the H-latch conferred resistance to prion infection. High-affinity ligands that prevented H-latch induction repressed prion-related neurodegeneration in organotypic cerebellar cultures. We then selected phage-displayed ligands binding wild-type PrP^C, but not PrP^2Cys. These binders depopulated H-latched conformers and conferred protection against prion toxicity. Finally, brain-specific expression of an antibody rationally designed to prevent H-latch formation prolonged the life of prion-infected mice despite unhampered prion propagation, confirming that the H-latch is an important reporter of prion neurotoxicity.

The neurotoxicity of prions requires the interaction of the misfolded prion protein PrP^Sc with its cellular counterpart PrP^C (ref. [1]), which ultimately leads to depletion of the PIKfyve kinase[2] and to spongiform encephalopathy. Prion toxicity is initiated by unknown mechanisms that require membrane-bound PrP^C (refs. [1,3]). PrP^C is a glycosylphosphaidylinositol (GPI)-anchored protein composed of an amino-terminal, unstructured 'flexible tail' (FT) and a carboxy-terminal, structured 'globular domain' (GD)[4]. Mice lacking the prion protein gene *Prnp* do not succumb to prion diseases[5]. Antibodies binding the globular domain (GD) of PrP^C can halt this process[6], but they can also activate toxic intracellular cascades[7–9]. Similar events occur in prion-infected brains, and substances that counteract the damage of infectious prions can also alleviate the toxicity of anti-PrP^C antibodies, such as POM1 (ref. [8]). POM1 exerts its toxicity without inducing the formation of infectious prions[10], arguing that toxicity is independent of prion replication. Accordingly, toxicity can be very effectively prevented by the therapeutic co-stabilization of FT and GD through bispecific antibodies[11]. These findings suggest that POM1 and prions exert their toxicity through similar mechanisms.

To explore the causal links between the binding of POM1 to PrP^C and its neurotoxic consequences, we performed structural and molecular studies in silico, in vitro and in vivo. We found that the induction of an intramolecular hydrogen bond between R208 and H140 of the globular domain of human PrP^C (hPrP^C) is an early molecular reporter of prion toxicity.

## Results

**POM1 introduces an intramolecular hydrogen bond in PrP^C-GD.** Structural analysis and molecular dynamics (MD) simulations indicated that POM1 induces an intramolecular hydrogen bond in both human and murine PrP^C between R208 and H139 in murine PrP^C (ref. [12]). This 'H-latch' constrains the POM1 epitope while allosterically increasing the flexibility of the β2–α2 and α2–α3 loops (Fig. 1 and Extended Data Fig. 1). To explore its role in prion toxicity, we generated a murine PrP^R207A mutant that prevents the H-latch without altering the conformation of PrP (Extended Data Fig. 1). We stably expressed murine PrP^R207A (mPrP^R207A) in *Prnp*^–/– CAD5 cells[13] and *Prnp*^ZH3/ZH3 cerebellar organotypic cultured slices (COCS; Fig. 2a–c and Extended Data Fig. 2a–c)[14,15]. A panel of conformation-specific anti-PrP antibodies showed similar staining patterns for PrP^C and mPrP^R207A, confirming that both proteins folded properly but had reduced POM1 binding (Extended Data Fig. 2d,e), as expected from the structure of PrP–POM1 co-crystals[12]. *Prnp*^–/– CAD5 cells expressing mPrP^R207A were resistant to POM1 toxicity and, notably, showed impaired prion replication (Fig. 2d–f), pointing to common toxic properties.

Lack of the H-latch confers resistance to prion and POM1 toxicity. To test whether its presence can induce toxicity even in the absence of ligands, we designed an R207C-I138C double-cysteine PrP^C mutant (PrP^2Cys Fig. 3a,b), with the goal of replicating the structural effects of the H-latch in the absence of POM1 binding. Nuclear magnetic resonance (NMR) and MD analysis of recombinant mPrP^2Cys were consistent with a folded protein resembling the

[1]Institute of Neuropathology, University of Zurich, Zurich, Switzerland. [2]Institute for Research in Biomedicine, Università della Svizzera italiana, Bellinzona, Switzerland. [3]PetMedix Ltd, Babraham Research Campus, Cambridge, UK. [4]Laboratory of Glia Biology, VIB-KU Leuven Center for Brain and Disease Research, Leuven, Belgium. [5]B23 Beamline, Diamond Light Source, Harwell Science Innovation Campus, Didcot, UK. [6]University of Zurich, Department of Chemistry, Zurich, Switzerland. [7]Department of Chemistry, University of Cambridge, Cambridge, UK. [8]Laboratory of Synapse Biology, Instituto de Investigação e Inovação em Saúde (i3S), University of Porto, Porto, Portugal. [9]These authors contributed equally: Karl Frontzek, Marco Bardelli, Assunta Senatore. [10]These authors contributed equally: Luca Varani, Adriano Aguzzi. ✉e-mail: luca.varani@irb.usi.ch; adriano.aguzzi@usz.ch

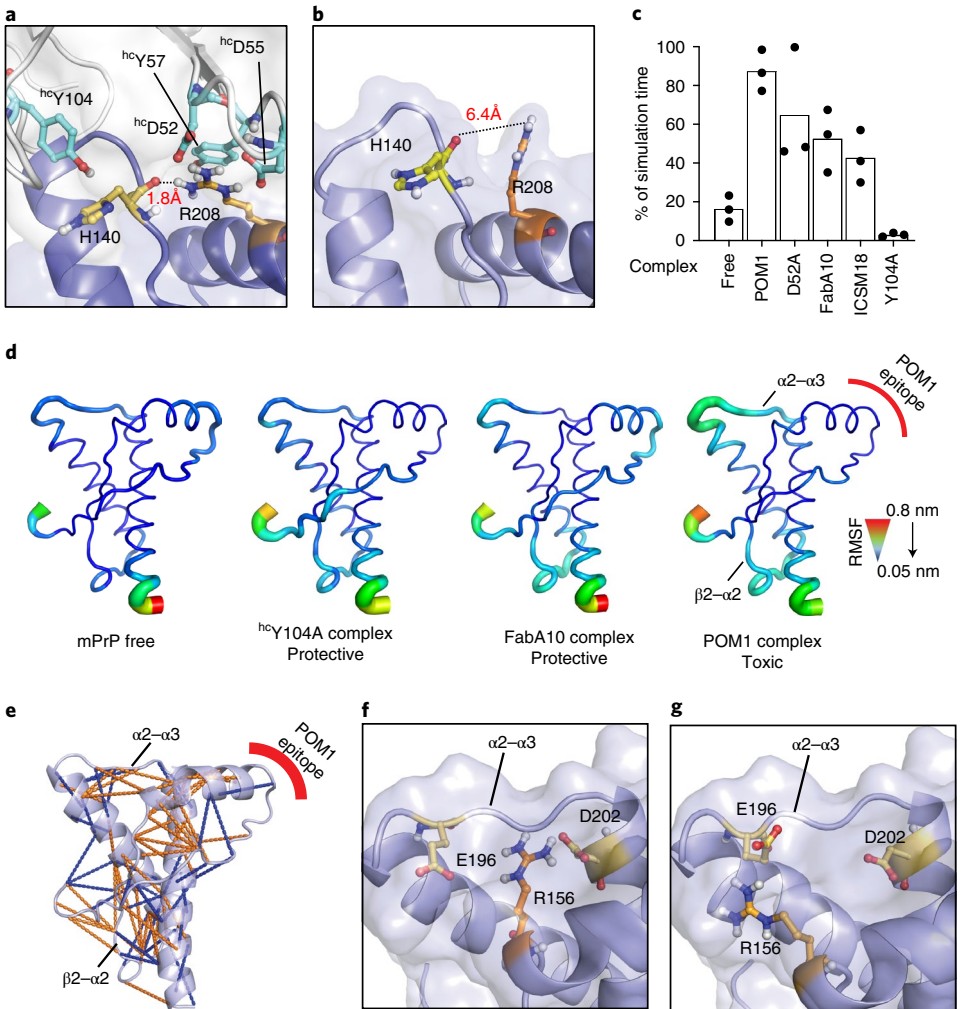

**Fig. 1 | POM1 induces an intramolecular hydrogen bond between R208A and H140 of human PrP^C. a,b**, Binding of PrP^C to the neurotoxic antibody POM1 favors the formation of a R208-H140 hydrogen bond in the GD of PrP^C (**a**) that is absent from free PrP^C (**b**). **c**, MD simulations indicate that toxic antibodies are more likely to induce the R208-H140 bond. Ordinate: percentage of simulation time in which the H-bond is present. See also Supplementary Figure 1. **d**, GD flexibility according to MD simulations. Narrow blue ribbons: rigidity; large green/red ribbons: increased flexibility. PrP bound to protective pomologs resembles free PrP. PrP bound to POM1 induces increased flexibility in the α2–α3 and β2–α2 loops. **e**, Binding of the toxic antibody POM1 to PrP induces local structural changes within the GD, here shown as a cartoon, both within and outside the epitope region. Side-chain contacts (less than 5 Å) that are present only in PrP free (blue, PDB 1xyx) or PrP bound (orange, PDB 4H88) are indicated by lines. **f**, POM1 binding breaks the R156-E196 interaction, increasing α2–α3 flexibility, and induces the formation of a R156-D202 salt bridge. **g**, R156 interacts with E196 in free PrP, which helps to rigidify the α2–α3 loop.

H-latch conformation (Fig. 3a–c). PrP^2Cys expressed in a *Prnp^−/−* CAD5 cell line showed correct glycosylation and topology and did not trigger unfolded protein responses (Extended Data Fig. 3b,c). Surface-bound PrP^2Cys was detected by POM8 and POM19, which bind to a conformational epitope on the the α1–α2 and β1–α3 regions, respectively[7], but not by POM1 (Extended Data Fig. 2d,e). The POM1-induced H-latch allosterically altered the β2–α2 loop; similarly, binding of mPrP^2Cys to POM5 (recognizing the β2–α2 loop[7]) was impaired (Extended Data Fig. 2a). Taken together, these results suggest that mPrP^2Cys adopts a conformation similar to that induced by POM1 (Fig. 3c). We transduced *Prnp^ZH3/ZH3* COCS with adeno-associated virus-based vectors (AAV) expressing either PrP^C or PrP^2Cys. Wild-type and mutant proteins showed similarly robust expression levels (Extended Data Fig. 3d). COCS expressing mPrP^2Cys developed spontaneous, dose-dependent neurodegeneration 4 weeks after transduction (Fig. 3d–f and Extended Data Fig. 3e,f), suggesting

that induction of the H-latch is sufficient to generate toxicity. In agreement with this view, MD simulations showed that human, hereditary PrP mutations responsible for fatal prion diseases favor H-latch formation and altered flexibility in the α2–α3 and β2–α2 loops (Extended Data Fig. 4).

**'Pomologs' rescue prion-induced neurodegeneration.** If POM1 toxicity requires the H-latch, antibody mutants that are unable to induce it should be innocuous. POM1 immobilizes R208 by salt bridges with its heavy-chain (hc) residue ^hcD52, whereas ^hcY104 contributes to the positioning of H140 (Fig. 1a). To prevent H-latch formation, we thus replaced eleven of these residues with alanine. For a control, we similarly substituted interface residues that are predicted to have no impact on R208. Resulting 'pomologs' were produced as single-chain variable fragments (scFv), three of which retained high affinity, that is a dissociation constant (K_D) of about 10 nM, for PrP^C (Table 1 and Extended Data Fig. 5).

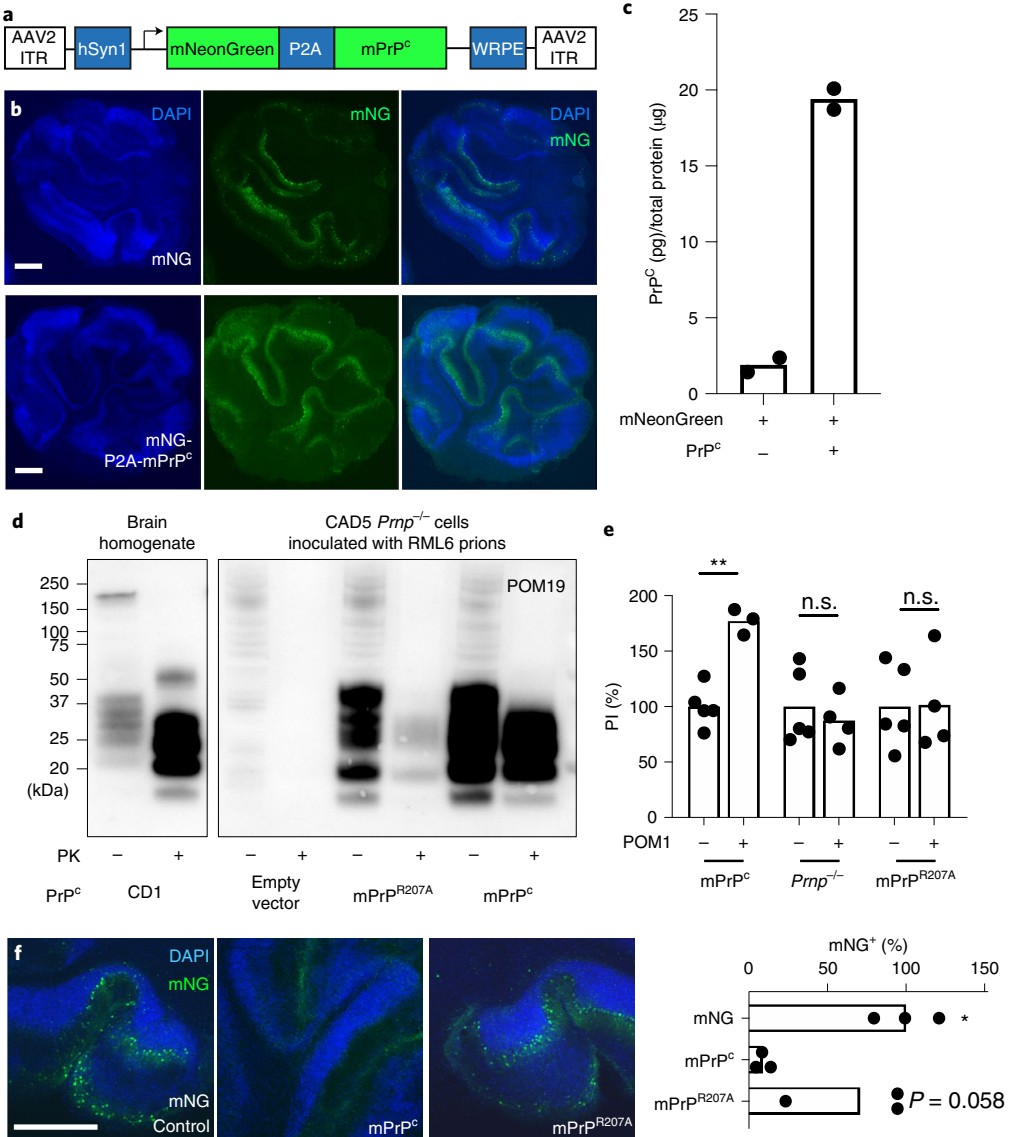

**Fig. 2 | Ablation of H-latch formation by a R207A mutation in murine PrP^C rescues PrP-induced toxicity. a**, Scheme of AAV used for bi-cistronic expression of monomeric NeonGreen and PrP^C, separated by a P2A site (monomeric neon green (mNG)-P2A-PrP^C). hSyn1, human Synapsin 1 promoter. WRPE, woodchuck hepatitis virus regulatory posttranscriptional element. ITR, inverted terminal repeats. **b**, Robust expression of mNG-P2A-PrP^C on fluorescent micrographs from transduced *Prnp*^ZH3/ZH3 COCS. Scale bars: 500 μm. **c**, Holo-POM19–holo-POM2-biotin PrP^C sandwich ELISA of samples depicted in **b**. One data point corresponds to a pool of 6–9 biological replicates of organotypic cultured slices. **d**, Proteinase K digestion of brain homogenates and cell lysates from chronically RML6-inoculated CAD5 cells (fourth passage is shown) detected with POM19. RML6 prions (lanes 1 and 2) and inoculated CAD5-mPrP^C cells (lanes 7 and 8) show a typical 'diagnostic shift' of proteinase K (PK)-digested PrP^Sc, whereas only trace amounts of PrP^Sc are detectable in CAD5-mPrP^R207A cells (lanes 5 and 6). Lack of detectable PrP^Sc in CAD5 *Prnp*^-/- (lanes 3 and 4) cells indicates no residual inoculum. Lanes are from non-adjacent samples blotted on the same membrane. **e**, Addition of POM1 causes toxicity to CAD5 cells (left) but not to *Prnp*^-/- or mPrP^R207A CAD5 cells (center and right). The percentage of propidium iodide (PI)-positive cells, determined by fluorescence-activated cell sorting (FACS), is shown on the *y* axis. Values are given as percentages of CAD5 mPrP^C PI-positive cells without POM1. One data point corresponds to a biologically independent cell lysate, for example a different cell passage. n.s., not significant, adjusted *P* > 0.05, **adjusted *P* = 0.0083, ordinary, one-way analysis of variance (ANOVA) with Šídák's multiple comparisons test. The FACS gating strategy is summarized in Extended Data Figure 3a. **f**, *Prnp*^ZH3/ZH3 COCS transduced with wild-type mPrP^C are susceptible to POM1 toxicity, whereas COCS transduced with control vector ('mNG control') or mPrP^R207A are not. Values are given as percentage of empty control. One data point corresponds to a biologically independent organotypic cultured slice. *adjusted *P* = 0.012, ordinary, one-way ANOVA with Šídák's multiple comparisons test. Scale bar: 500 μm.

As expected, all pomologs were innocuous to *Prnp*^ZH1/ZH1 COCS not expressing PrP^C (ref. [5]) (Extended Data Fig. 6a and Supplementary Fig. 1a). ^hcY104A reduced H-latch formation, according to MD simulations (Fig. 1b and Supplementary Fig. 2) and exerted no neurotoxicity onto COCS from tga20 mice over-expressing PrP^C (ref. [16]), whereas POM1 and all H-latch inducing

mutants (^hcD52A, ^hcY101A and all light-chain pomologs) were neu-rotoxic (Fig. 4a and Extended Data Fig. 6b). As with POM1, the tox-icity of pomologs required PrP^C, featured neuronal loss, astrogliosis and elevated levels of microglia markers (Extended Data Fig. 6c and Supplementary Fig. 1b), and was ablated by co-administration of the antibody POM2, which targets the flexible tail (FT) of PrP^C

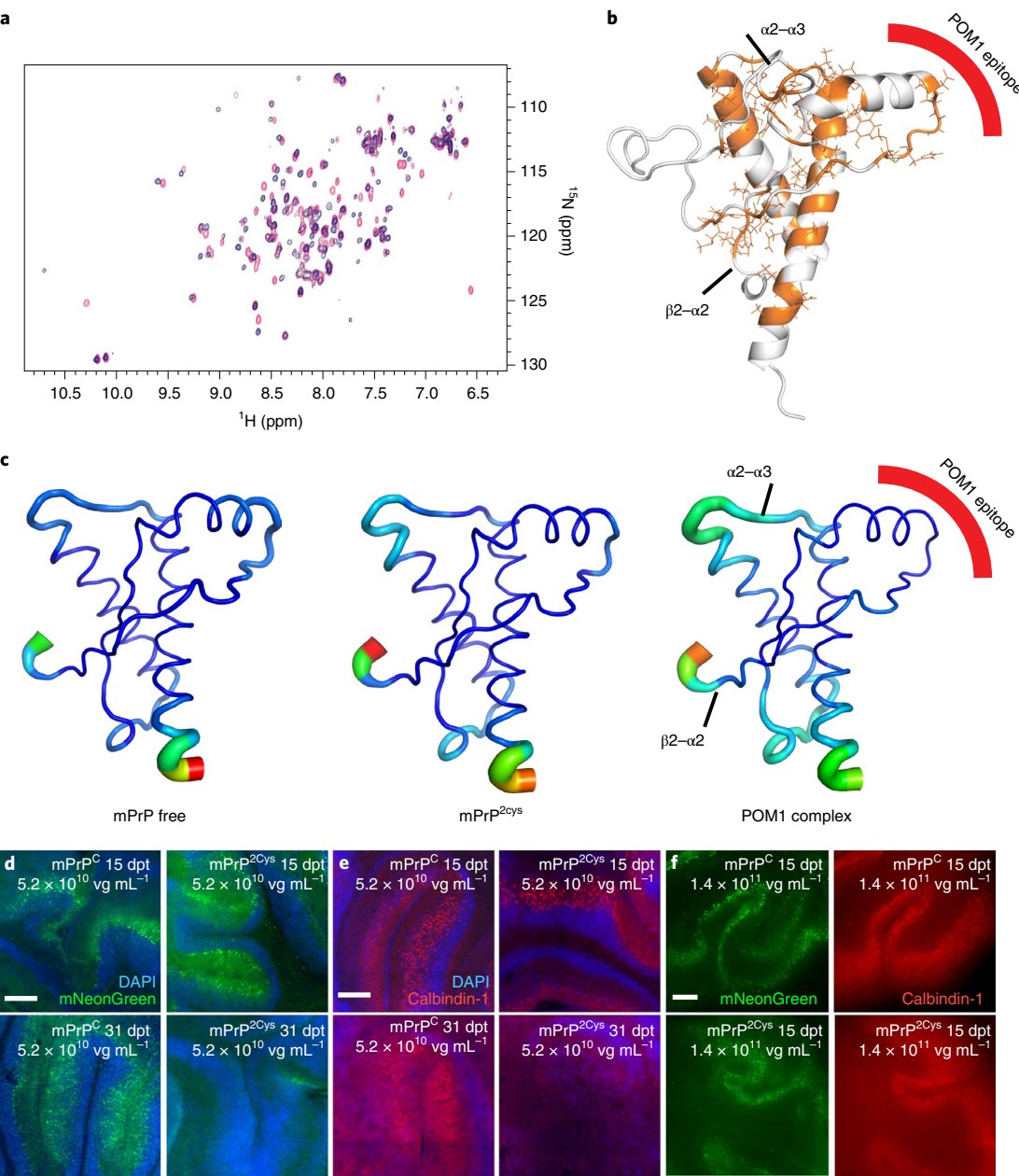

**Fig. 3 | The R207C–I138C double-cysteine PrP$^C$ mutant acts as an H-latch mimic. a,b**, $^{15}$N-heteronuclear single quantum coherence spectra of rmPrP free (red) and mPrP$^{2cys}$ (blue). Residues with different chemical shifts in the two spectra are colored orange on the GD structure in **b**, which resemble the H-latch conformation in the POM1–PrP complex. **c**, MD simulations show that mPrP$^{2cys}$ resembles the PrP–POM1 complex, with increased flexibility in the α2–α3 and β2–α2 loops and decreased flexibility in the 2Cys region, corresponding to the POM1 epitope. **d–f**, Prnp$^{ZH3/ZH3}$ COCS transduced with a bi-cistronic AAV expressing mNG and mPrP$^C$ (left) or mPrP$^{2Cys}$ (right). See Extended Data Figure 3f for quantification. Scale bars: 250 μm. **d**, mNG was visible in all COCS at 15 days post transduction (dpt, top row) but disappeared in mPrP$^{2Cys}$ at 31 dpt (bottom row). **e**, Calbindin-1$^+$ Purkinje cells were preserved at 15 dpt but became largely undetectable at 31 dpt, possibly as a result of mPrP$^{2Cys}$ toxicity. **f**, Dose escalation of twice as many viral vectors as in **d** and **e** led to earlier onset of mPrP$^{2Cys}$-mediated neurodegeneration. Significant neurodegeneration was observable at 15 dpt; see quantification in Extended Data Figure 3f.

(Extended Data Fig. 6d)[7]. Additionally, $^{hc}$Y104A inhibited POM1 toxicity (Extended Data Fig. 6e,f).

POM1 does not induce de novo prions[10] but triggers similar neurotoxic cascades[8], plausibly by replicating the docking of prions to PrP$^C$. If so, $^{hc}$Y104A may prevent the neurotoxicity of both POM1 and prions by competing for their interaction with PrP$^C$. Indeed, $^{hc}$Y104A protected RML6 and 22L prion-inoculated tga20 and C57BL/6 COCS from prion neurodegeneration (Fig. 4b–d and Extended Data Fig. 6g–i), repressed the vacuolation of chronically prion-infected cells (Fig. 4e and ref. [2]) and diminished PrP$^{Sc}$ levels ex vivo (Fig. 4f). In contrast to other antiprion antibodies[17], $^{hc}$Y104A did not reduce levels of PrP$^C$ (Fig. 4g), corroborating the conjecture that neuroprotection results from interfering with the docking of incoming prions.

The antibody ICSM18 was found to ameliorate prion toxicity in vivo[18], although dose-escalation studies have shown conspicuous neuronal loss[9]. The ICSM18 epitope is close to that of POM1 (ref. [12]), and MD simulations indicated that it facilitates the R208-H140 interaction, albeit less so than POM1 does (Fig. 1c).

**Table 1 | a,** Computational alanine scanning indicates which residues of POM1 and PrP contribute to binding. Positive numbers in the third column suggest loss of binding energy. **b,** On the basis of these results (Table 1a), we prepared 11 single mutations of POM1 (in each CDR loop) as scFv constructs. Colors (yellow to red) visualize the impact on binding affinity. The mutated residues are shown as sticks on the cartoon POM1 structure in Extended Data Figure 5b

**A**

| Res N° | chain | ΔG(complex) | |
|--------|-------|-------------|---|
| 32 | L | 0.44 | |
| 50 | L | 1.96 | |
| 91 | L | 2.09 | |
| 92 | L | 0.27 | |
| 93 | L | 0.28 | POM1 residues |
| 94 | L | 1.39 | |
| 96 | L | 0.35 | |
| 33 | H | 2.88 | |
| 52 | H | 2.47 | |
| 54 | H | -0.02 | |
| 55 | H | 1.37 | |
| 57 | H | 1.76 | |
| 59 | H | -0.01 | |
| 101 | H | 0.59 | |
| 103 | H | -0.03 | |
| 104 | H | 4.91 | |
| 139 | A | -0.01 | |
| 146 | A | -0.01 | |
| 138 | A | 0.11 | |
| 143 | A | -0.21 | |
| 145 | A | 0.31 | |
| 141 | A | 0.39 | PrP residues |
| 212 | A | 0.78 | |
| 140 | A | 0.93 | |
| 147 | A | 1.9 | |
| 144 | A | 3.44 | |
| 208 | A | 3.86 | |

**B**

| | $k_a$ (1/Ms) | $k_d$ (1/s) | $K_D$ (nM) | |
|---|---|---|---|---|
| **POM1** | $6.4 \times 10^4$ | $3.1 \times 10^{-4}$ | 4.8 | |
| **hcW33A** | no binding | no binding | no binding | CDR_H1 loop |
| **hcD52A** | $3.3 \times 10^5$ | $4.6 \times 10^{-2}$ | 103 | |
| **hcD55A** | $1 \times 10^5$ | $3.7 \times 10^{-2}$ | 372 | CDR_H2 loop |
| **hcY57A** | $7.7 \times 10^4$ | $3.1 \times 10^{-2}$ | 406 | |
| **hcY101A** | $6.3 \times 10^4$ | $7.1 \times 10^{-4}$ | 11 | |
| **hcY104A** | $3.6 \times 10^4$ | $2.1 \times 10^{-4}$ | 8.8 | CDR_H3 loop |
| **lcS32A** | $6.4 \times 10^4$ | $5.9 \times 10^{-4}$ | 10 | CDR_L1 loop |
| **lcY50A** | $5.7 \times 10^4$ | $8.2 \times 10^{-3}$ | 313 | CDR_L2 loop |
| **lcS91A** | $1.3 \times 10^4$ | $1.8 \times 10^{-2}$ | 1430 | |
| **lcW94A** | $3.1 \times 10^5$ | $2.7 \times 10^{-2}$ | 201 | CDR_L3 loop |
| **lcY96A** | $6.2 \times 10^4$ | $7.4 \times 10^{-3}$ | 123 | |

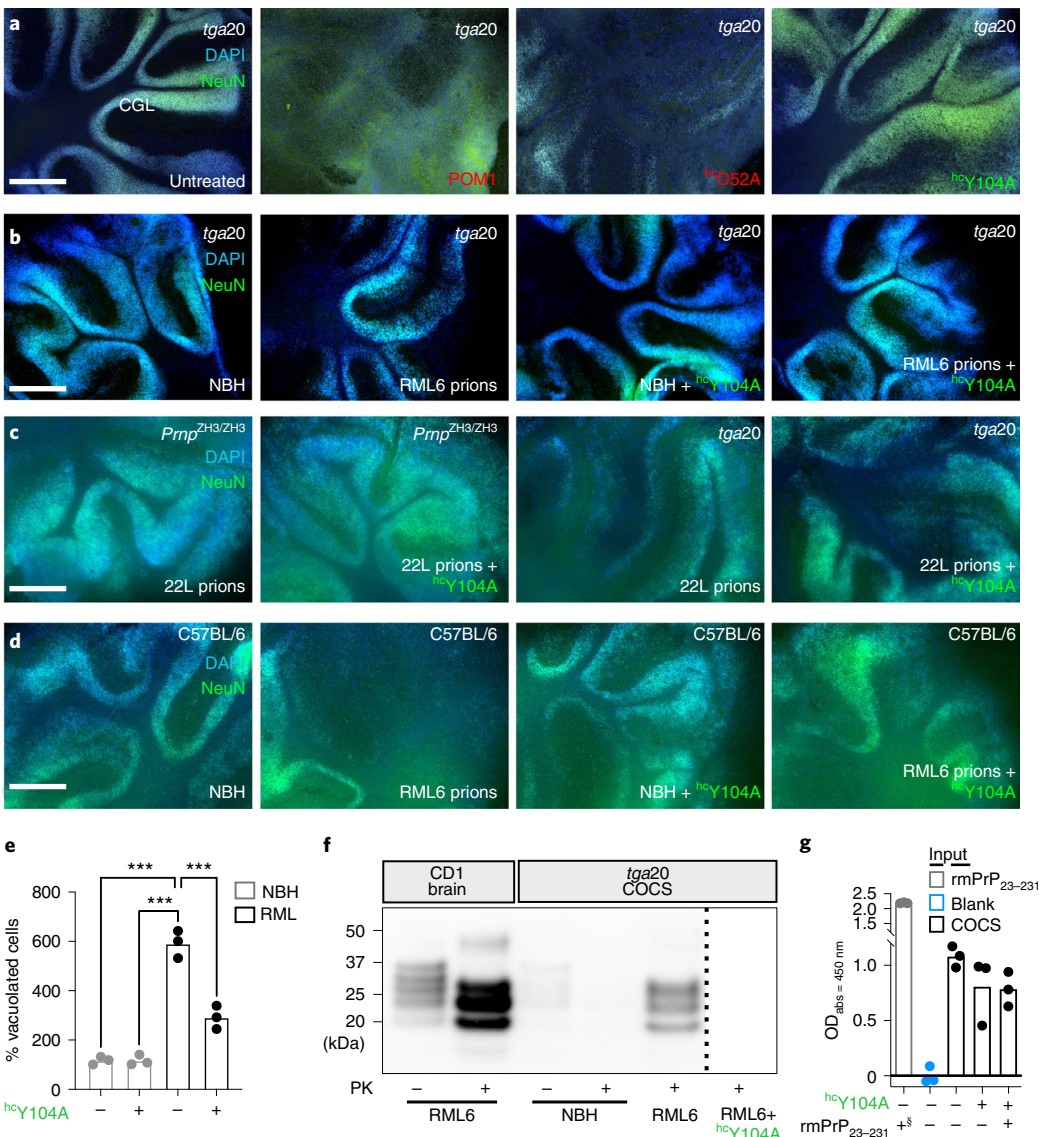

**Fig. 4 | Preventing H-latch formation by pomologs rescues prion-induced neurodegeneration. a**, The densely cellular NeuN⁺DAPI⁺ cerebellar granule cell layer (CGL) of tga20 COCS was preserved by treatment with POM1 mutant [hc]Y104A (green) but destroyed by POM1 and [hc]D52A (red). **b**, CGL degeneration occurs in prion-infected tga20 COCS, but not in COCS exposed to non-infectious brain homogenate (NBH). Treatment of RML6 prion-infected tga20 COCS with [hc]Y104A prevented neuronal loss. **c**, Rescue of prion-induced toxicity by [hc]Y104A in COCS inoculated with 22L prions. **d**, Treatment of prion-infected wild-type COCS, expressing wild-type levels of PrP[C], with [hc]Y104A prevented CGL degeneration. **a–d**, Quantification of fluorescent micrographs is depicted in Extended Data Figure 6b,g–i. Scale bar: 500 µm. **e**, Treatment with [hc]Y104A (180 nM; 5 days) reduced vacuolation in chronically prion-infected Gt1 cells. Each dot represents an independent experiment with cells from different passages (1,000 cells/experiment, ordinary one-way ANOVA with Dunnett's multiple comparisons test, ****adjusted $P < 0.0001$). **f**, Treatment of prion-infected tga20 COCS with [hc]Y104A led to a reduction in PrP[Sc] levels. One lane corresponds to a pool of 6–9 COCS digested with PK; PrP[Sc] was detected using holo-POM1. The dashed bar indicates gel splicing of lanes running in non-adjacent wells on the same gel. **g**, Treatment of tga20 COCS with [hc]Y104A for 7 days did not reduce PrP[C] levels, as determined by PrP[C] sandwich ELISA. §870 pM of rmPrP₂₃₂₃₀ were used as a positive control (first lane). Pomologs were pre-incubated with 600 nM of rmPrP₂₃₋₂₃₀ as negative controls (last lane). Ordinate: absorbance, given as optical density at $\lambda = 450$ nm.

**Antibody binding causes conformational changes in GD and FT.** Protective pomolog [hc]Y104A failed to induce the H-latch, which was induced by toxic mutations (Fig. 1c and Extended Data Fig. 1). MD simulations showed that POM1 rigidified its epitope but increased the flexibility of the α2–α3 and β2–α2 loops (Fig. 1c). Conversely, the conformation of PrP attached to the protective [hc]Y104A resembled that of free PrP. In accordance with MD simulations, NMR spectra, which are sensitive to local effects and transient populations[19], of rmPrP₉₀₋₂₃₁ in complex with POM1 revealed long-range alterations in the GD and in the adjacent FT (Fig. 5a). When bound to [hc]Y104A instead, rmPrP₉₀₋₂₃₁ elicited spectra similar to those of free PrP. Circular-dichroism (CD) spectroscopy showed that the full rmPrP (rmPrP₂₃₋₂₃₁)–POM1 complex had more irregular structure content than its free components (Fig. 5b), whereas no difference was observed when POM1 was complexed to partially FT-deficient rmPrP₉₀₋₂₃₁. We did not observe any changes in the secondary structure of the [hc]Y104A-bound rmPrP₂₃₋₂₃₁ complex. This suggests that POM1 can alter the FT conformation with two possible mechanisms. Either the secondary structure of the FT itself is changed, probably through a shift in the population of conformers (FT-changes), or the

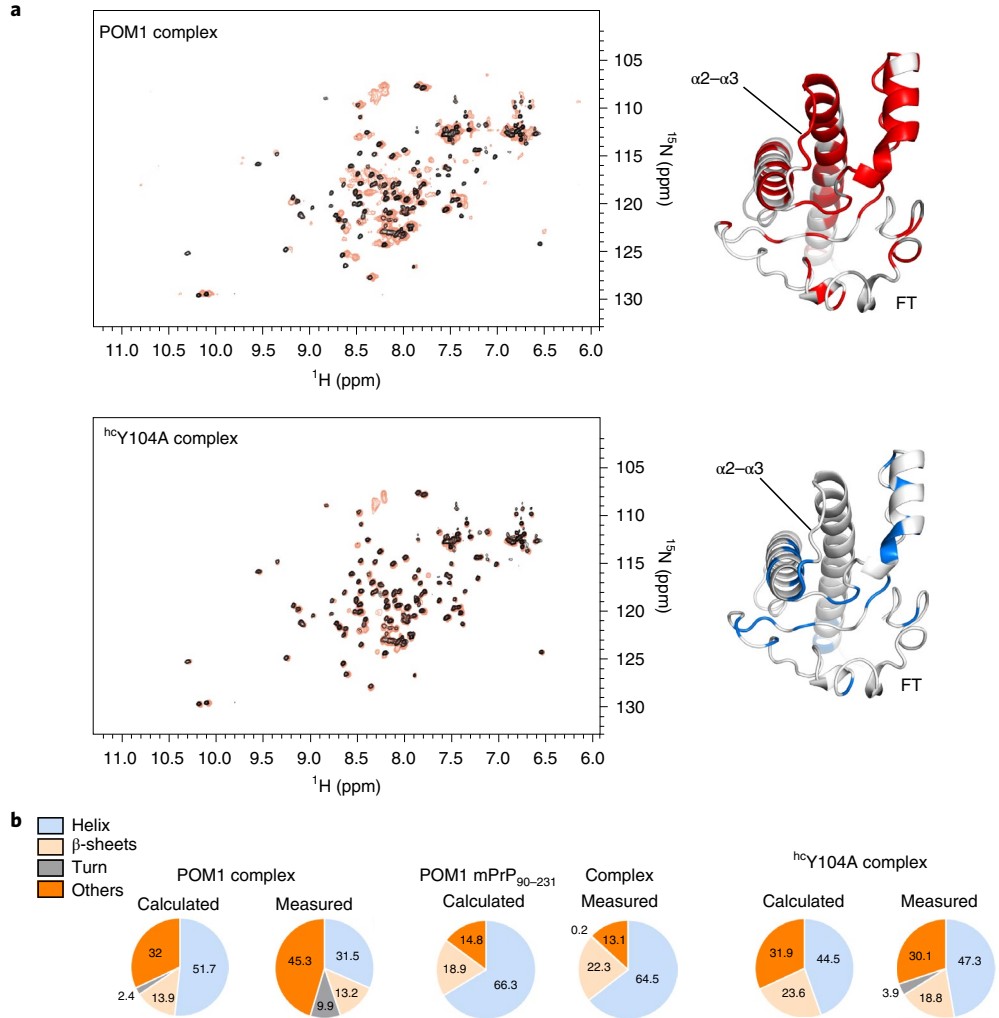

**Fig. 5 | Antibody binding causes allosteric conformational changes in the GD and FT. a**, Comparison between the [$^{15}$N,$^1$H]-TROSY spectra of free rmPrP$_{90-231}$ versus that bound to the $^{hc}$Y104A pomolog. Chemical-shift differences, reflecting subtle alterations of the local chemical structure, were visible not only in the epitope but also at distant sites in the GD and FT. Residues affected by antibody binding are in color on PrP$^C$ (GD and part of the FT are shown on a MD model of PrP). Differences between toxic and protective antibodies are evident in the α2–α3 loop (the Y104A complex is identical to free PrP$^C$) and in the FT region closer to the GD. **b**, Content of secondary structure estimated from CD spectra of the rmPrP–pomologs complexes. 'Calculated' indicates the secondary structure content if the rmPrP and pomolog did not change upon binding. POM1 displayed increased content of irregular structure (measured versus calculated) when in complex with full rmPrP$_{23-231}$, but identical content when in complex with a construct lacking the FT (rmPrP$_{90-231}$). This indicates that the FT changes conformation upon POM1 binding. Conversely, no differences were detected with the protective pomolog $^{hc}$Y104A.

secondary structure of the GD is altered in a FT-dependent manner, with FT-GD interactions stimulated by POM1 binding. Hence H-latch induction leads to subtle alterations of the structure of both GD and FT, whose presence correlates with toxicity.

We performed animal experiments to confirm that (1) $^{hc}$Y104A by itself is not neurotoxic in vivo, in contrast to POM1, and (2) it protects from prion-dependent neurodegeneration. When produced as IgG holoantibody, $^{hc}$Y104A exhibited subnanomolar affinity to full-length, murine, recombinant PrP (rmPrP$_{23-231}$, Supplementary Fig. 2). We injected POM1 or holo-$^{hc}$Y104A into the hippocampus of C57BL/6 mice. Histology and volumetric-diffusion-weighted magnetic resonance imaging showed that POM1 (6 µg) elicited massive neurodegeneration that was repressed by pre-incubation with recPrP in threefold molar excess, whereas the same amount of holo-$^{hc}$Y104A did not elicit any tissue damage (Fig. 6a–g and Extended Data Figs. 7 and 8). A benchmark dose analysis[9] yielded an upper safe-dose limit of ≥12 µg for intracerebrally injected holo-$^{hc}$Y104A (Extended Data Fig. 8a). Also, the injection of holo-$^{hc}$Y104A (6 µg) into tga20 mice,

which are highly sensitive to POM1 damage, failed to induce any lesions (Extended Data Fig. 8b–e).

We then transduced tga20 mice with $^{hc}$Y104A by intravenous injection of a neurotropic AAV-PHP.B vector. Two weeks after AAV injection, mice were inoculated intracerebrally with $3 \times 10^5$ ID$_{50}$ units of RML6 prions. $^{hc}$Y104A expression levels correlated with both survival times and PrP$^{Sc}$ deposition (Fig. 6h–j), suggesting that $^{hc}$Y104A acts downstream of prion replication.

**Phage displayed antibody fragments confer neuroprotection.** If the same toxic PrP conformation is induced by both the H-latch and infectious prions, anti-PrP antibodies unable to bind the H-latch conformers could depopulate them by locking PrP$^C$ in its innocuous state, thus preventing prion neurotoxicity. Using phage display (Extended Data Fig. 9a), we generated four antigen-binding fragments (Fabs), three of which bound the globular domain of PrP$^C$ preferentially over PrP$^{2Cys}$, with one binding PrP and PrP$^{2Cys}$ similarly (Fig. 7a and Extended Data Fig. 9b). When administered to

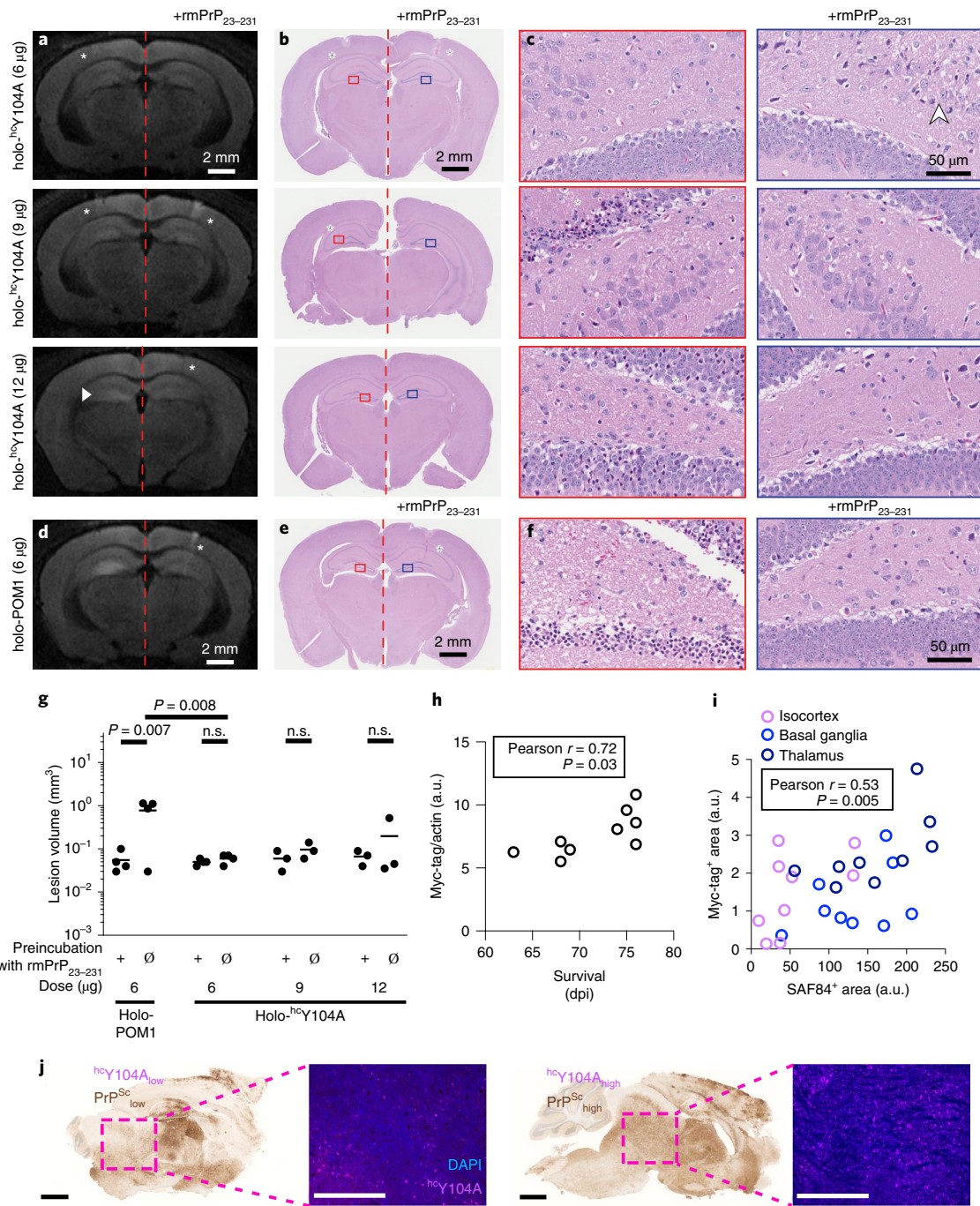

**Fig. 6 | The holo-IgG antibody ^hcY104A is innocuous after intracerebral injection. a**, Representative magnetic resonance diffusion-weighted images (DWI) 24 hours after stereotactic injection of holo-^hcY104A (left). Contralateral injections of holo-^hcY104A + rmPrP$_{23231}$ (right). A small area of hyperintensity was found in one mouse after injection of 12 μg holo-^hcY104A (white arrowhead). White asterisks: needle tract. **b**, Hematoxylin and eosin (HE)-stained sections from mice shown in **a**. Asterisks: needle tract. Rectangles denote regions magnified in **c**. **c**, HE sections (CA4). Left, holo-^hcY104A injections (6, 9 and 12 μg). Right, holo-^hcY104A + rmPrP$_{23-230}$. Asterisk (9 μg): neurons with hypereosinophilic cytoplasm and nuclear condensation in the vicinity of the needle tract. Asterisk (12 μg): These neurons were diffusely distributed among numerous healthy neurons. White arrowhead: vacuoles indicative of edema along the needle tract. **d**, DWI images of 6 μg holo-POM1 ± rmPrP$_{23-231}$, revealing a hyperintense signal at 24 hours. **e**, HE-stained section from a mouse shown in **d**. Asterisks: needle tract. Rectangles: areas in **f**. **f**, HE sections (CA4). Holo-POM1 injections revealed damaged neurons with condensed chromatin and hypereosinophilc cytoplasm. **g**, Volumetric quantification of lesions on DWI imaging 24 hours after injection revealed no significant lesion induction by holo-^hcY104A. One datapoint corresponds to an animal. P values are adjusted for multiple comparisons. n.s.: not significant, $P > 0.05$, ordinary one-way ANOVA with Šídák's multiple comparisons test. **h**, Antibody expression levels, as determined by Myc-Tag western blot, showed a positive correlation with survival. One datapoint corresponds to one animal. Pearson correlation coefficient $r = 0.72$, 95% confidence interval 0.099–0.94, $P = 0.03$. a.u., arbitrary units. **i**, Significant correlation of PrP$^{Sc}$ and antibody expression levels (representative images depicted in **j**, aggregated correlation across all brain regions). Different colors represent 3 brain regions from 9 independent animals. Pearson correlation coefficient $r = 0.53$, 95% confidence interval 0.18–0.76, $P = 0.0048$. **j**, Representative images from quantification of **l**. **l**, Sagittal brain sections stained with SAF84, highlighting PrP$^{Sc}$, and basal ganglia immunofluorescent micrographs marking ^hcY104A-Myc-tag. Scale bar SAF84: 1mm. Scale bar ^hcY104A-Myc-tag: 500 μm.

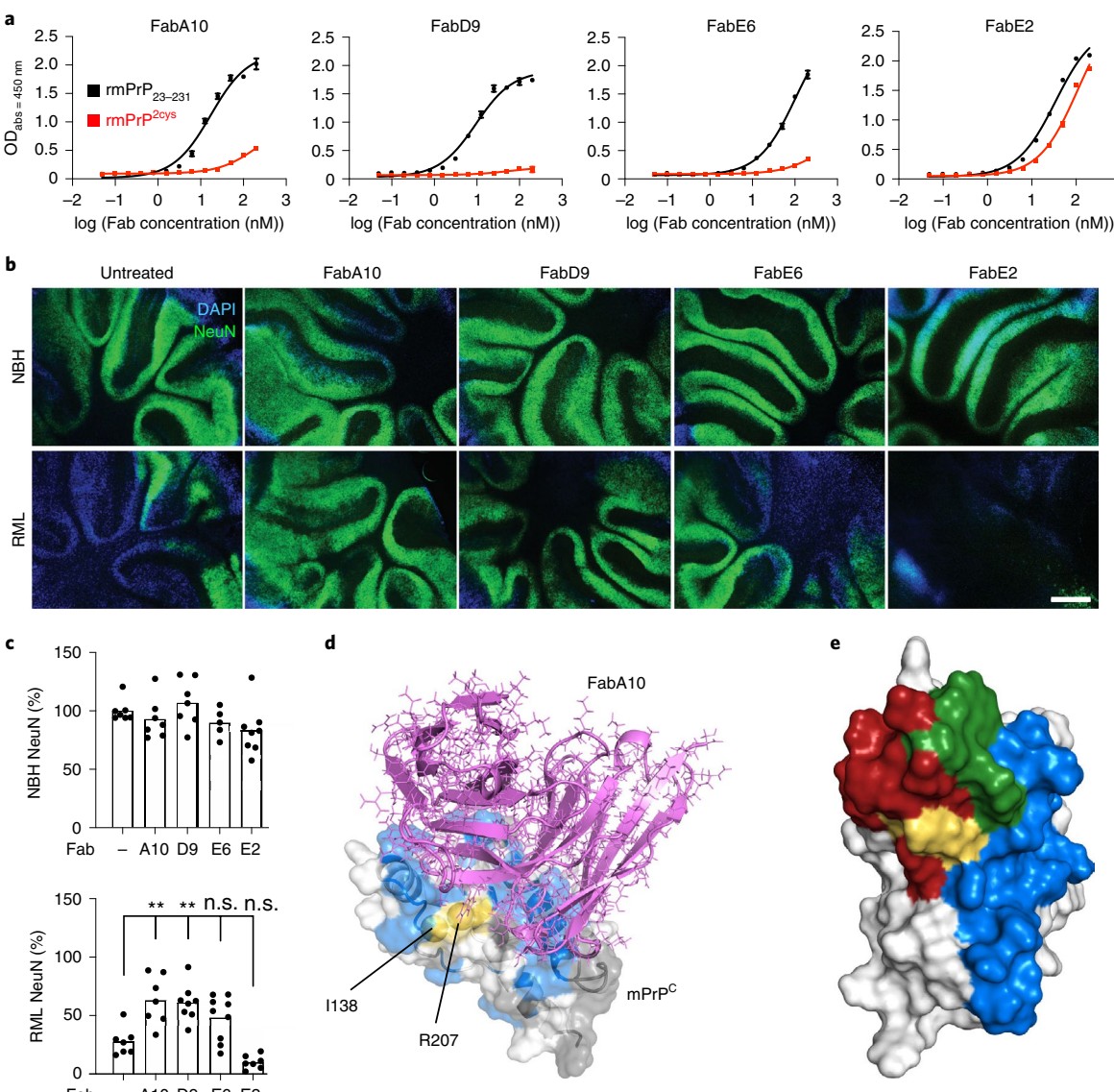

**Fig. 7 | Phage-displayed antibody fragments differentially binding wild-type PrP^C, but not PrP^2Cys, confer neuroprotection. a**, Preferential binding of the selected Fabs to rmPrP$_{23-231}$ over rmPrP^2Cys. With the exception of FabE2, the Fabs show higher apparent affinity for rmPrP$_{23-231}$ than rmPrP^2Cys. One datapoint corresponds to the mean ± s.e.m. of two technical replicates. The experiment was repeated twice. **b**, FabA10 and FabD9 conferred neuroprotection in prion-infected tga20 COCS. **c**, Quantification of NeuN fluorescence intensity from **b**, expressed as percentage of untreated (−) NBH. Scale bar: 500 μm. One datapoint corresponds to an independent, organotypic cultured slice. Two-way ANOVA with Dunnett's multiple comparison test, *P* values are adjusted for multiple testing: RML untreated (−) versus RML A10: *P* = 0.006, RML untreated (−) versus RML D9: *P* = 0.009, **P* < 0.01, n.s.: not significant, *P* > 0.05. **d**, Structure of PrP^C (white) in complex with FabA10 (violet) obtained by NMR-validated docking and MD. mPrP$_{90-231}$ residues whose NMR signal is affected by FabA10 binding are colored blue; residues with no NMR information are gray; residues mutated to Cys are yellow. **e**, There is partial overlap (green) between the epitopes of POM1 (red) and FabA10 (blue). The 2Cys are in yellow. PrP^C is depicted in different orientations in **d** and **e**.

prion-infected tga20 COCS, FabA10 and FabD9 decreased prion neurotoxicity, whereas FabE2, which binds both PrP^C and mPrP^2Cys, had no beneficial effect (Fig. 7b,c). NMR epitope mapping followed by computational docking and MD[20] showed that FabA10 binds to PrP encompassing the H-latch and partially overlapping with the POM1 epitope (Fig. 7d,e and Extended Data Fig. 10). MD showed that the H-latch is not stable in the presence of FabA10, even if simulations were started from a POM1-bound PrP conformation with the R208-H140 H-bond present (Extended Data Fig. 10a).

## Discussion

In summary, the evidence presented here suggests that H-latch formation is an important feature of prion toxicity. The H-latch was induced by the toxic anti-PrP antibody POM1, PrP mutants unable to form the H-latch conferred resistance to POM1 toxicity, and a PrP mutant mimicking the H-latch was constitutively neurotoxic. Conversely, POM1 mutants retaining their affinity and epitope specificity, but abolishing H-latch formation, proved to be neuroprotective. We observed that formation of the H-latch and its structural effects on PrP^C-GD were not only innocuous, but also protective against prion neurotoxicity in vitro and in vivo. The MD predictions were confirmed in vivo using both cerebellar slice cultures and mouse models of prion disease. POM1 mutants or other rationally selected Fabs that were unable to induce the H-latch protected from the deleterious effects of prion infection ex vivo and in vivo. Furthermore, hereditary PrP mutations leading to human prion

diseases favor the H-latch, according to MD simulations. These observations suggest that the H-latch is not only involved in the toxicity of anti-PrP antibodies, but also in the pathogenesis of prion diseases.

Spongiform change, that is endolysosomal hypertrophy through UPR activation and subsequent PIKfyve depletion, is shared in both prion and POM1 toxicity[21]. Multiple toxic cascades are activated in prion infections and in cells treated with POM1 (ref. [8]). Cells that stably express PrP[2cys] are not affected by UPR in the current experimental paradigm, suggesting that either the protein dosage is insufficient to observe UPR or its toxicity is independent of PIKfyve depletion. Besides neuronal loss, which is shared among prion, POM1 and PrP[2cys] toxicity, it will be interesting to investigate the overlap of toxic cascades between the different prion disease models, which could provide important knowledge of early disease-associated changes.

The above findings hold promise for therapeutic interventions. First, the POM1 binding region includes a well-defined pocket created by the α1–α3 helix of PrP[C], which may be targeted by therapeutic compounds including antibodies, small molecules, cyclic peptides or aptamers. Second, [hc]Y104A halted progression of prion toxicity even when it was already conspicuous, whereas the anti-FT antibody POM2 exerted neuroprotection only when applied directly after prion inoculation[11]. This suggests that [hc]Y104A halts prion toxicity upstream of FT engagement[8,11]. Thirdly, tga20 COCS (which are much more responsive to toxic pomologs than wild-type COCS, and can therefore be regarded as a sensitive sentinel system) tolerated prolonged application of [hc]Y104A at concentrations around $150 \times K_D$. Finally, intracerebrally injected [hc]Y104A was innocuous, and AAV-transduced [hc]Y104A extended the lifespan of prion-infected mice, despite elevated PrP[Sc] levels, suggesting that it acts downstream of PrP[Sc] replication, possibly by blocking a PrP[Sc]-PrP[C] interaction at the POM1 epitope. These findings suggest that blockade of the POM1 epitope by agents that do not induce the H-latch has good in vivo tolerability. In view of the reports that PrP[C] may mediate the toxicity of disparate amyloids[22], the relevance of the above findings may extend to proteotoxic diseases beyond spongiform encephalopathies.

## Online content

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

## Methods

**Adeno-associated virus production and in vivo transduction.** Single-stranded adeno-associated virus (ssAAV) vector backbones with AAV2 inverted terminal repeats (ITRs) were kindly provided by B. Schneider (EPFL). Herein, expression of the monomeric NeonGreen fluorophore was driven by the human Synapsin I (hSynI) promoter. A P2A sequence (GSGATNFSLLKQAGDVEENPGP) was introduced between mNG and PrP$^C$ for bi-cistronic expression. For mPrP$^{R207A}$ and mPrP$^{2cys}$ expression, a synthetic gene block (gBlock, IDT, full sequence is given in the Supplementary Information) was cloned between the BsrGI and HindIII site of the vector replacing the wild-type PrP$^C$ sequence. Recombination of plasmids was tested using SmaI digestion prior to virus production. The viral vectors and viral vector plasmids were produced as hybrid AAV2/6 (AAV6 capsid with AAV2 ITRs) by the Viral Vector Facility (VVF) of the Neuroscience Center Zurich. The identity of the packaged genomes was confirmed by Sanger DNA-sequencing (identity check). Quantification of mNG-positive cells from confocal images was done using the Spots function in Imaris (Bitplane).

Neurotropic AAV variants for scFv antibody expression were constructed from a synthetic gene fragment, NheI-IL2-scFv-Myc-EcoRV (produced by Genscript Biotech), which contained $^{hc}$Y104A sequences preceded by the signal peptide from interleukin-2 (IL-2)[23]. NheI and EcoRV restriction-enzyme digestion was performed on NheI-IL2-scFv-Myc-EcoRV synthetic gene fragments, which were then inserted into a ssAAV vector backbone. ScFv expression was under the control of the strong, ubiquitously active CAG promoter. A WPRE (woodchuck hepatitis virus post-transcriptional regulatory element) sequence was also included, downstream of the transgene, to enhance transgene expression. Production, quality control and determination of vector titer was performed by ViGene Biosciences). Rep2 and CapPHP.B plasmids were provided under a Material Transfer Agreement (MTA). Further details about packaging and purification strategies can be found on the company's website (http://www.vigenebio.com).

**Allen Mouse Brain Atlas data.** Images from in situ hybridization for calbindin 1 and synapsin 1 expression were taken from the Allen Mouse Brain Atlas (www.brain-map.org). The first dataset retrieved by the R package allenbrain (https://github.com/oganm/allenBrain) with the closest atlas image to the center of the region (regionID = 512, settings: planeOfSection = 'coronal', probeOrientation = 'antisense') was downloaded (dataset ID nos. for calb1 = 71717640, syn1 = 227540).

**Animals and in vivo experiments.** We conducted all animal experiments in strict accordance with the Swiss Animal Protection law and dispositions of the Swiss Federal Office of Food Safety and Animal Welfare (BLV). The Animal Welfare Committee of the Canton of Zurich approved all animal protocols and experiments performed in this study (animal permits 123, ZH90/2013, ZH120/16, ZH139/16). Genetically modified mice from the following genotypes were used in this study: Zurich I mice homozygous for disrupted *Prnp* genes (*Prnp*$^{o/o}$, denoted as *Prnp*$^{ZH1/ZH1}$)[5], Zurich III *Prnp*$^{o/o}$ (denoted as *Prnp*$^{ZH3/ZH3}$)[15] and tga20 (ref. [16]).

For in vivo transduction with the neurotropic AAV-PHP.B construct, mice received a total volume of 100 μL (1 × 10$^{11}$ total vector genomes) by intravenous injection into the tail vein. Fourteen days after AAV transduction, the left hemispheres of Tga20 mice were inoculated with 30 μL of 0.1% RML6 brain homogenate, corresponding to 3 × 10$^5$ median lethal dose (LD$_{50}$; 3.6 μg of total brain homogenate). Brain homogenates were prepared in 0.32 M sucrose in PBS at a concentration of 10% (wt/vol). Protein analysis of mouse brains is described below.

After fixation with 4% paraformaldehyde for 1 week, tissues were treated with concentrated formic acid for 60 minutes, fixed again in formalin and eventually embedded in paraffin. HE staining and SAF84 immunohistochemistry were performed as previously described[24]. For immunohistochemical detection of Myc-tag, tissue was deparaffinized and incubated in citrate buffer (pH 6.0) in a domestic microwave for 20 minutes. Unspecific reactivity was blocked using blocking buffer (10% goat serum, 1% bovine serum albumin, 0.1% Triton X-100 in PBS) for 1 hour at room temperature. Primary rabbit anti-Myc-tag antibody (1:500, ab9106, Abcam, overnight at 4 °C) was detected with Alexa Fluor 594 rabbit anti-goat (IgG) secondary antibody (1:1,000, 1 hour at room temperature), diluted in staining buffer (1% bovine serum albumin, 0.1% Triton X-100 in PBS). Tissue was counterstained with DAPI (5 μg/ml, 15 minutes at room temperature).

**Cell lines.** CAD5 is a subclone of the central nervous system catecholaminergic cell line CAD showing particular susceptibility to prion infection[10]. Generation of the CAD5 *Prnp*$^{-/-}$ clone no. C12 was described before, as was overexpression of murine, full-length PrP$^C$ in CAD5 *Prnp*$^{-/-}$ by cloning the open reading frame of *Prnp* into the pcDNA3.1(+) vector, *Prnp* expression was driven by a constitutively expressed cytomegalovirus promoter (yielding pcDNA3.1(+)-*Prnp*) as described earlier[13]. For stable expression of mPrP$^{2cys}$, pcDNA3.1(+)-*Prnp* vector was modified using Quikchange II Site-Directed Mutagenesis Kit (Agilent), according to the manufacturer's guidelines. We first introduced a mutation leading to p.R207C (primers (5′→3′): mutagenesis forward: GTG-AAG-ATG-ATG-GAG-TGC-GTG-GTG-GAG-CAG-A, reverse: TCT-GCT-CCA-CCA-CGC-ACT-CCA-TCA-TCT-TCA-C) which was then followed by the p.I138C mutation (mutagenesis forward: AGT-CGT-TGC-CAA-AAT-GGC-ACA-TGG-GCC-TGC-TCA-TGG,

reverse: CCA-TGA-GCA-GGC-CCA-TGT-GCC-ATT-TTG-GCA-ACG-ACT). For stable expression of mPrP$^{R207A}$, pcDNA3.1(+)-*Prnp* was mutated correspondingly (mutagenesis forward: TGT-GAA-GAT-GAT-GGA-GGC-CGT-GGT-GGA-GCA-GAT-G, reverse: TCT-GCT-CCA-CCA-CGC-ACT-CCA-TCA-TCT-TCA-C).

**Cell vacuolation assay.** Mouse hypothalamic Gt1 neuronal cells were grown in Dulbecco's Modified Eagle Medium (DMEM) in the presence of 10% fetal bovine serum (FBS), 1% penicillin–streptomycin and 1% glutamax (all obtained from Invitrogen). For prion infection of the cells, Gt1 cells grown in DMEM were incubated with either Rocky mountain laboratory strain of prion (RML6) prions (0.1%) or non-infectious brain homogenate (NBH; 0.1%) for 3 days in 1 well of a 6-well plate. This was followed by splitting the cells at a 1:3 ratio every 3 days for at least 10 passages. The presence of infectivity in the cells was monitored by the presence of PK-resistant PrP, as described below. At 70 days postinfection (dpi), the cells started developing vacuoles, which were visualized by phase-contrast microscopy. Antibody treatment with $^{hc}$Y104A was administered on 70–75 dpi at a concentration of 180 nM.

**Cerebellar organotypic slice cultures.** Mice from C57BL/6, tga20, *Prnp*$^{ZH1/ZH1}$ and *Prnp*$^{ZH3/ZH3}$ strains were used for preparation of COCS, as described[14]. Herein, 350-μm-thick COCS were prepared from 9- to 12-day-old pups. Free-floating sections of COCS were infected with 100 μg prions per 10 slices of RML6 (passage 6 of the Rocky Mountain Laboratory strain mouse-adapted scrapie prions) or 22L (mouse-adapted scrapie prions) brain homogenate from terminally sick, prion-infected mice. Brain homogenate from CD1-inoculated mice was used as non-infectious brain homogenate (NBH). Sections were incubated with brain homogenates diluted in physiological Grey's Balanced Salt Solution 1 hour at 4 °C and washed, and 5–10 slices were placed on a 6-well PTFE membrane insert. Analogously, for AAV experiments, COCS were incubated with AAV at a final concentration of 5.2 × 10$^{10}$ total vector genomes diluted in physiological Grey's Balanced Salt Solution for 1 hour at 4 °C, then washed and placed on PTFE membrane inserts. Antibody treatments were given thrice weekly, e.g. with every medium change. In naive slices, antibody treatments were initiated after a recovery period of 10–14 days.

For testing of innocuity of pomologs (Fig. 2c, Supplementary Fig. 7c and Supplementary Fig. 10), POM1 and pomolog antibodies were added at 400 nM for 14 days. Supplementary Figures 7c and 10 represent aggregated data from multiple experiments with COCS from mice of identical genotype and age; compounds were administered at identical timepoints and dosage. When added to RML-infected tga20 COCS (Fig. 2d and Supplementary Fig. 7d), $^{hc}$Y57A was added from 20 to 45 dpi, $^{hc}$Y104A was added from 21 to 45 dpi and both antibodies were given at 400 nM. Antibody treatment with $^{hc}$Y57A and $^{hc}$Y104A of RML-infected tga20 COCS used for determination of PrP$^{Sc}$ by western blot, see detailed protocol below, was initiated and stopped at 21 dpi and 45 dpi, respectively. $^{hc}$D55A was added to RML-infected tga20 COCS at either 1 (800 nM, Supplementary Fig. 12d) or 21 (400 nM, Fig. 2d and Supplementary Fig. 7d) dpi. When added to C57BL/6 COCS (Fig. 2e and Supplementary Fig. 7e), hc$^{Y104A}$ was added from 1 dpi at 400 nM until 45 dpi. In 22L-inoculated COCS, $^{hc}$Y104A was administered at 21 dpi, and slices were collected at 44 dpi. Phage-derived Fabs were added to RML-infected COCS (Fig. 4b–g) from 1 dpi until 45 dpi at 550 nM.

**ELISA.** PrP$^C$ levels were measured by ELISA using monoclonal anti-PrP$^C$ antibody pairs POM19/POM3 or POM3/POM2 (all as holo-antibodies), as described previously[25]. First, 384-well SpectraPlates (Perkin Elmer) were coated with 400 ng mL$^{-1}$ POM19 (POM3) in PBS at 4 °C overnight. Plates were washed 3 times in 0.1% PBS-Tween 20 (PBS-T) and blocked with 80 μL 5% skim milk in 0.1% PBS-T per well for 1.5 hours at room temperature. Blocking buffer was discarded, and samples and controls were dissolved in 1% skim milk in 0.1% PBS-T for 1 hour at 37 °C. Twofold dilutions of rmPrP$_{23–231}$, starting at a dilution of 100 ng/mL in 1% skim milk in 0.1% PBS-T, were used as a calibration curve. Biotinylated POM3 (POM2) was used to detect PrP$^C$ (200 ng/mL in 1% skim milk in 0.1% PBS-T), and biotinylated antibodies were detected with streptavidin-HRP (1:1,000 in 1% skim milk in 0.1% PBS-T, BD Biosciences). Chromogenic reaction and reading of plates were performed as described in ref. [25]. Unknown PrP$^C$ concentrations were interpolated from the linear range of the calibration curve using linear regression (GraphPad Prism, GraphPad Software).

**ELISA screening of phage display.** Single colonies were picked and cultured in a 384-well plate (Nunc) in 2YT, ampicillin and 1% glucose medium overnight at 37 °C, 80% humidity, 500 r.p.m. These precultures were used to prepare glycerol stock master plates. Expression plates were prepared from the master plates by inoculating corresponding wells with 2YT, carbenicillin and 0.1% glucose medium, followed by induction with 1 mM IPTG. After 4 hours at 37 °C, 80% humidity, cultures were lysed for 1.5 hours at 400 r.p.m., 22 °C in borate-buffered saline, pH 8.2, containing EDTA-free protease inhibitor cocktail, 2.5 mg/mL lysozyme and 40 U/mL benzonase. Fab-containing bacteria lysate was blocked with Superblock and used for ELISA screening, and the reactivity to four different antigens was assessed in parallel. The following antigens were coated on separate

384-well ELISA plates: anti-Fd antibody (The Binding Site) 1:1,000 in PBS, to check the expression level of each Fab clone in bacteria; rmPrP$_{23–231}$ at 87 nM in PBS, to identify candidate PrP$^C$ binders; mPrP$^{2cys}$ at 87 nM in PBS, to check for cross reactivity with mPrP$^{2cys}$; neutravidin at 87 nM as a control for specificity. Antigen-coated ELISA plates were washed twice with PBS-T and blocked with Superblock for 2 hours. Fab-containing bacteria lysates from the expression plate were transferred to corresponding wells of the ELISA plates. After 2 hours of incubation, ELISA plates were washed 3 times with PBS-T, and anti-human-F(ab′)2-alkaline-phosphatase-conjugated antibody (1:5,000 in PBS-T) was added. After 1 hour of incubation at room temperature, followed by 3 washings with PBS-T, pNPP substrate was added and, after 5 min incubation, the ELISA signal was measured at 405 nm. Fabs from bacteria lysates producing an ELISA signal 5 times higher than the technical background, which was calculated as the average of the coated well containing un-inoculated medium, and negative for neutravidin were considered as PrP$^C$ binder candidates. For hit selection, we considered only anti-PrP$^C$ Fabs whose ELISA signal for rmPrP$_{23–231}$ was at least two times higher than for mPrP$^{2cys}$. All the identified hits were checked in a confirmatory ELISA screening. Bacterial cultures of the selected clones were used for DNA minipreps followed by Sanger sequencing using the following sequencing primers: HuCAL_VH (5′-GATAAGCATGCGTAGGAGAAA-3′) and M13Rev (5′-CAGGAAACAGCTATGAC-3′).

**Expression and purification of selected anti-PrP Fabs.** Chemically competent BL21(D3) cells (Invitrogen) were transformed with selected pPE2-Fab plasmids and grown on plates with LBagar, kanamycin and 1% glucose. A single colony was inoculated into 20 mL of 2×YT, kanamycin and 1% glucose pre-culture medium and incubated for at least 4 hours at 37 °C, 220 r.p.m. One liter of 2YT medium containing kanamycin and 0.1% glucose was inoculated with 20 mL pre-culture, and Fab expression was induced by 0.75 mM IPTG followed by incubation overnight at 25 °C, 180 r.p.m. The overnight culture was centrifuged at 4,000g at 4 °C for 30 minutes, and the pellet was frozen at −20 °C. For Fab purification, the thawed pellet was resuspended into 20 mL lysis buffer (0.025 M Tris pH 8; 0.5 M NaCl; 2 mM MgCl$_2$; 100 U/mL benzonase (Merck); 0.25 mg/mL lysozyme (Roche), EDTA-free protease inhibitor (Roche)) and incubated for 1 hour at room temperature at 50 r.p.m. The lysate was centrifuged at 16,000g at 4 °C for 30 minutes, and supernatant was filtrated through 0.22-μM Millipore Express Plus Membrane. Fab purification was achieved via the His$_6$-Tag of the heavy chain by IMAC. Briefly, after equilibration of the Ni-NTA column with running buffer (20 mM Na-phosphate buffer, 500 mM NaCl, 10 mM imidazole, pH 7.4), and the bacteria lysate was loaded and washed with washing buffer (20 mM Na-phosphate buffer, 500 mM NaCl, 20 mM Imidazole, pH 7.4). The Fab was eluted with elution buffer (20 mM Na-phosphate buffer, 500 mM NaCl, 250 mM imidazole, pH 7.4). Buffer exchange was performed using PD-10 columns, Sephadex G-25M (Sigma), whereby the Fab was eluted with PBS.

**Förster resonance energy transfer.** Europium (Eu$^{3+}$) donor fluorophore was coupled to POM1 (yielding POM1-Eu$^{3+}$) and allophycocyanin (APC) acceptor fluorophores was coupled to holoantibody POM3 (yielding holo-POM3-APC) as previously described[26]. Full-length, recombinant mouse prion protein (rmPrP$_{23–231}$) was added at a final concentration of 1.75 nM, followed by addition of holo-POM3-APC at a final concentration of 5 nM and subsequent incubation at 37 °C for 30 minutes with constant shaking at 400 r.p.m. Pomologs were then added in serial dilutions from 0 to 3 nM and were again incubated at 37 °C for 60 minutes with constant shaking at 400 r.p.m., followed by addition of POM1-Eu$^{3+}$ at a final concentration of 2.5 nM. Net Förster resonance energy transfer (FRET) was calculated as described previously[26].

**Determination of binding constants from Förster resonance energy transfer.** The dependence of the FRET signal on POM1 concentration was modelled by a simple competitive binding model. The binding constant of the FRET-labeled POM1-Eu$^{3+}$ was defined as:

$$K_F = \frac{[PrP_{free}] \times [F_{free}]}{[F_b]} = \frac{([PrP_{tot}] - [F_b] - [A_b]) \times ([F_{tot}] - [F_b])}{[F_b]}$$

where square brackets denote concentration, $F_{tot}$, $F_{free}$ and $F_b$ denote total, free and bound POM1-Eu$^{3+}$, $PrP_{tot}$ and $PrP_{free}$ denote the total and free PrP, $A_{tot}$, $A_{free}$ and $A_b$ denote total, free and bound single-chain fragment variables (scFvs) and $K_F$ is the binding constant of POM1-Eu$^{3+}$. The righthand equality is obtained by imposing conservation of mass. An equivalent equation defines the binding constant of the scFvs:

$$K_D = \frac{[PrP_{free}] \times [A_{free}]}{[A_b]} = \frac{([PrP_{tot}] - [F_b] - [A_b]) \times ([A_{tot}] - [A_b])}{[A_b]}$$

This system of equations is solved to give $F_b$ as a function of $A_{tot}$. To relate the concentration of bound POM1-Eu$^{3+}$, $F_b$, to the FRET measurements, this equation was rescaled to 100 for the fully bound and 10 for the fully unbound limit. An additional complication in interpreting the experimental data stems from the fact that a FRET signal will appear only if both a POM1-Eu$^{3+}$and holo-POM3-APC

are bound to the same PrP. We assume that the binding of POM1 and POM3 is independent, so we can approximate the concentration of PrP bound to a holo-POM3-APC as the effective PrP concentration, $PrP_{tot}$ in the above equations. The binding constant of holo-POM3-APC was determined to be 0.23 nM, giving an effective concentration of PrP of 1.64 nM (compared with the total PrP concentration of 1.75 nM). To verify the robustness of these results, we fitted the data assuming a much weaker binding of holo-POM3-APC, with a binding constant of 1 nM. The obtained $K_D$ values of the single-chain fragments were within the error of the ones determined with a holo-POM3-APC binding constant of 0.23 nM.

**Immunohistochemical stainings and analysis of immunofluorescence.** COCS were washed twice in PBS and fixed in 4% paraformaldehyde for at least 2 days at 4 °C and were washed again twice in PBS prior to blocking of unspecific binding by incubation in blocking buffer (0.05% Triton X-100 vol/vol, 0.3% goat serum vol/vol in PBS) for 1 hour at room temperature. For visualization of neuronal nuclei, the monoclonal mouse anti-NeuN antibody conjugated to Alexa Fluor 488 (clone A60, Life Technologies) was dissolved at a concentration of 1.6 μg mL$^{-1}$ into blocking buffer and incubated for 3 days at 4 °C. Further primary antibodies used were recombinant anti-calbindin antibody (1 μg mL$^{-1}$, ab108404, Abcam), anti-glial fibrillary acidic protein (1:500, Z0334, DAKO) and anti-F4/80 (1 μg mL$^{-1}$, MCAP497G, Serotec). Unconjugated antibodies were dissolved in blocking buffer and incubated for 3 days at 4 °C. After 3 washes with PBS for 30 minutes, COCS were incubated for 3 days at 4 °C with secondary antibodies Alexa-Flour-594-conjugated goat anti-rabbit-IgG (Life Technologies) or Alexa-Fluor-647-conjugated goat anti-rat-IgG (Life Technologies) at a dilution of 1:1,000 in blocking buffer. Slices were then washed with PBS for 15 minutes and incubated in DAPI (1 μg mL$^{-1}$) in PBS at room temperature for 30 minutes to visualize cell nuclei. Two subsequent washes in PBS were performed, and COCS were mounted with fluorescence mounting medium (DAKO) on glass slides. NeuN, GFAP, F4/80 and calbindin morphometry was performed by image acquisition on a fluorescence microscope (BX-61, Olympus), and analysis was performed using gray-level auto thresholding function in ImageJ (www.fiji.sc). Cell numbers in Figure 2f were determined using the 'Spots' function in Imaris (Oxford Instruments). Morphometric quantification was done on unprocessed images with identical exposure times and image thresholds between compared groups. Representative fluorescent micrographs in the main and supplementary figures have been processed (linear adjustment of brightness and contrast) for better interpretability.

For immunohistochemistry of CAD5 cells, cells were seeded on 18-well μ-slides (Ibidi) and fixed with 4% paraformaldehyde for 5 minutes at room temperature. Unspecific reactions were blocked using 3% goat serum in PBS for 1 hour at room temperature. Mouse monoclonal anti-PrP$^C$ antibodies POM1, POM5, POM8 and POM19 (all holo-antibodies) were established before[25]; POM antibodies were incubated at 4 μg mL$^{-1}$ in 3% goat serum in PBS at 4 °C, followed by 3 washes in PBS. Antibodies were detected using Alexa-Fluor-488-conjugated goat anti-mouse-IgG at 1:250 dilution, followed by nuclear counterstain with DAPI (1 μg mL$^{-1}$ in PBS) for 5 minutes at room temperature. Image analysis was performed using SP5 confocal microscope (Leica) with identical exposure times across different experimental groups.

**In vitro toxicity assessment.** Quantification of POM1 toxicity on CAD5 Prnp$^{-/-}$ cells stably transfected with mPrP$^C$, mPrP$^C_{R207A}$ or empty control vector, as described above, was measured as percentage of PI-positive cells using flow cytometry, as described before[11].

CAD5 cells were cultured with 20 mL Corning Basal Cell Culture Liquid Media–DMEM and Ham's F-12, 50/50 Mix, supplemented with 10% FBS, Gibco MEM Non-Essential Amino Acids Solution 1×, Gibco GlutaMAX Supplement 1× and 0.5 mg/mL of Geneticin in T75 Flasks (Thermo Fisher) at 37 °C, 5% CO$_2$. Sixteen hours before treatment, cells were split into 96-well plates at 25,000 cells/well in 100 μL.

POM1 alone was prepared at 5 μM final concentration in 20 mM HEPES pH 7.2 and 150 mM NaCl, and 100 μL of each sample, including buffer control, was added to CAD5 cells, in duplicates.

After 48 hours, cells were washed 2 times with 100 μL MACS buffer (PBS + 1% FBS + 2 mM EDTA) and resuspended in 100 μL MACS buffer. Thirty minutes before FACS measurements, PI (1 μg/mL) was added to the cells. Measurements were performed using a BD LSRFortessa. The percentages of PI-positive cells were plotted in columns as mean with s.d. The gating strategy is depicted in Extended Data Figure 3a.

**In vivo toxicity assessment.** The in vivo toxicity assessment was performed as previously described[9]. In brief, mice where i.c. injected using a motorized stereotaxic frame (Neurostar) at the following bregma coordinates (AP −2 mm, ML ±1.7 mm, DV 2.2mm, angle in ML/DV plane 15°). Antibodies (2 μL) were injected at a flow rate of 0.5 μL/minute. After termination of the injection, the needle was left in place for 3 minutes.

Twenty-four hours after stereotactic injection, mice were placed on a bed equipped with a mouse whole-body radio frequency transmitter coil and a mouse head surface-coil receiver and then transferred into the 4.7 Bruker Pharma scan. For DWI, routine gradient echo sequences with the following parameters were

used: TR: 300 ms TE: 28 ms, flip angle: 90°, average: 1, matrix: 350 × 350, field of view: 3 × 3 cm, acquisition time: 17 minutes, voxel size: 87 × 87 μm$^3$, slice thickness: 700 μm, isodistance: 1,400 μm$^3$ and $b$ values: 13,816 s/mm$^2$. Finally, mice were euthanized after 49 hours, and the brains were fixed in 4% formalin. Coronal sections from the posterior cortex were paraffin-embedded (4 mm) and 2-μm coronal step sections (standard every 100 μm) were cut, deparaffinized and routinely stained with hematoxylin and eosin.

Dose–response analysis and the benchmark dose relation were calculated with benchmark dose software (BMDS) 2.4 (United States Environmental Protection Agency).

**Molecular dynamics.** Experimental structures were used as a basis for MD simulations when available (scPOM1–mPrP complex, Protein Data Bank (PDB) 4H88; free mPrP, PDB 1XYX). The structures of full-length mPrP, mPrP$_{\Delta90–231}$ and the pomologs were predicted by homology modeling on the I-Tasser webserver[27], on the basis of the experimental structure of the PrP globular domain (aa 120–231), and were further validated with MD.

In all simulations, the system was initially set up and equilibrated through standard MD protocols: proteins were centered in a triclinic box, 0.2 nm from the edge, and filled with SPCE water model and 0.15 M Na$^+$Cl$^-$ ions using the AMBER99SB-ILDN protein force field; energy minimization followed. Temperature (298 K) and pressure (1 bar) equilibration steps (100 ps each) were performed. Three independent replicates of 500-ns MD simulations were run with the above-mentioned force field for each protein or complex. MD trajectory files were analyzed after removal of periodic boundary conditions. The overall stability of each simulated complex was verified by root mean square deviation, radius of gyration and visual analysis, according to standard procedures. Structural clusters, atomic interactions and root mean square fluctuation (RMSF) were analyzed using GROMACS[28] and standard structural biology tools. RMSF provides a qualitative indication of residue level flexibility, as shown in Figure 1c.

The presence of H-bonds or other interactions between GD residues was initially estimated by visual analysis and then by distance between appropriate chemical groups during the simulation time.

**Nuclear magnetic resonance.** Spectra were recorded on a Bruker Avance 600 MHz NMR spectrometer at 298 K, pH 7 in 50 mM sodium phosphate buffer at a concentration of 300 μM. In mapping experiments, mPrP was uniformly labeled with $^{15}$N (99%) and $^2$H (approx. 70%); antibodies were unlabeled. PrP and antibody samples were freshly prepared and extensively dialyzed against the same buffer prior to complex formation. The same procedure was followed for CD measurements. Chemical-shift assignment was based on published data (Biological Magnetic Resonance Data Bank entry 16071)[29]. Briefly, overlay of [$^{15}$N,$^1$H]-TROSY spectra of free or bound mPrP$_{90–231}$ allowed for identification of PrP residues for which the associated NMR signal changed upon complex formation, indicating alterations in their local chemical environment[19].

**Phage display.** A synthetic human Fab phagemid library (Novartis Institutes for BioMedical Research) was used for phage display. First, two rounds of selection against PrP$^C$ were performed by coating 96-well Maxisorp plates (Nunc) with a decreasing amount of rmPrP$_{23–231}$ (1 μM and 0.5 μM respectively, in PBS), overnight at 4 °C. PrP-coated plates were washed 3 times with PBS-T and blocked with Superblock for 2 hours. Input of 4 × 10$^{11}$ phages in 300 μL PBS was used for the first round of panning. After 2 hours of blocking with Chemiblocker (Millipore), the phages were incubated with PrP-coated wells for 2 hours at room temperature. The non-binding phages were then removed by extensive washing with PBS-T, while rmPrP$_{23–231}$-bound phages were eluted with 0.1 M glycine/HCl, pH 2.0 for 10 minutes at room temperature; the pH was then neutralized by 1 M Tris pH 8.0. Eluted phages were used to infect exponentially growing amber suppressor TG1 cells (Lubio Science). Infected bacteria were cultured in 2YT, carbenicillin and 1% glucose medium overnight at 37 °C, 200 r.p.m., and superinfected with VCSM13 helper phages. The production of phage particles was then induced by culturing the superinfected bacteria in 2YT, carbenicillin and kanamycin medium containing 0.25 mM isopropyl β-D-1-thiogalactopyranoside (IPTG), overnight at 22 °C, 180 r.p.m. Supernatant containing phages from the overnight culture was used for the second panning round. Output phages from the second round were purified by PEG/NaCl precipitation, titrated and used in the following third rounds to enrich phage-displayed Fabs that bound preferentially mPrP$^C$ over mPrP$^{2cys}$.

Two strategies were used: depletion of binders to recombinant mPrP$^{2cys}$ by subtraction in solid phase and depletion of mPrP$^{2cys}$ binders by competition with rhPrP$^C_{23–230}$-AviTag in liquid phase. In the former setting, purified phages were first exposed to 0.75 μM mPrP$^{2cys}$ (threefold molar excess compared with rmPrP$_{90–231}$ or rmPrP$_{121–231}$), and then the unbound phages were selected for rmPrP$_{90–231}$ or rmPrP$_{121–231}$ binders. Alternatively, purified phages were first adsorbed on neutravidin-coated wells to remove the neutravidin binders and then exposed to 0.25 μM rhPrP$^C_{23–230}$-AviTag in solution in the presence of 0.75 μM (threefold molar excess) of mPrP$^{2cys}$. The phage-displayed Fabs binding to rhPrP$^C_{23–230}$-AviTag were captured on neutravidin-coated wells and eluted as described above. For both strategies, a fourth panning round was performed using 0.3 μM mPrP$^{2cys}$ for depletion and 0.1 μM rmPrP$_{121–231}$ (coated on the plate) or rhPrP$^C_{23–230}$-AviTag

(in solution) for positive selection. At the fourth round of selection, DNA minipreps were prepared from the panning output pools by QIAprep Spin Miniprep kit (Qiagen) and the whole anti-PrP Fab enriched library was subcloned in expression vector pPE2 (kindly provided by Novartis). DNA was then used to transform electrocompetent non-amber suppressor MC1061 bacteria (Lubio Science) to produce soluble Fabs and perform ELISA screening.

**Production of recombinant proteins and antibodies.** Bacterial production of recombinant, full-length mouse PrP$_{23–231}$, recombinant fragments of human and mouse PrP and recombinant, biotinylated human PrP$^C$-AviTag (rhPrP$^C_{23–230}$-AviTag) was achieved as previously described[30–32]. Production of scFv and the IgG POM1 antibodies used in this manuscript was performed as described before[25]. Production of holo-$^{hc}$Y104A was performed as follows: POM1 IgG$_1$ heavy chain containing a Y104A mutation and POM1 kappa light chain were ordered as a bicistronic synthetic DNA block (gBlock, IDT) separated by a P2A site. The synthetic gene block (gBlock, IDT, see full sequence in the Supplementary Information) was then cloned into pcDNA 3.4-TOPO vector (Thermo Fisher Scientific), and recombinant expression was achieved using the FreeStyle MAX 293 Expression System (Thermo Fisher Scientific), according to the manufacturer's guidelines. Glucose levels were kept constant over 25 mM. Seven days after cell transfection, medium supernatant was collected, centrifuged and filtered. A Protein-G column (GE Healthcare) was used for affinity purification of antibodies, followed by elution with glycine buffer (pH 2.6) and subsequent dialysis against PBS (pH 7.2–7.4). Purity was determined by SDS–PAGE, and protein concentrations were determined using Pierce BCA Protein Assay Kit (Thermo Fisher Scientific). For generation of POM1 mutants, we performed site-directed mutagenesis on a POM1 pET-22b(+) (Novagen) expression plasmid[7] according to the manufacturer's guidelines (primers (5′→3′): $^{hc}$W33A: forward (FW): CATTCA CTGACTACGCGATGCACTGGGTGAAGC, reverse (REV): GCTTCACCCAG TGCATCGCGTAGTCAGTGAATG. $^{hc}$D52A: FW: GAGTGGATCGGATCGATT GCGCCTTCTGATAG, REV: CTATCAGAAGGCGCAATCGATCCGATCCACTC. $^{hc}$D55A: FW GGATCGATTGATCCTTCTGCGAGTTATACTAGTCAC, REVGT GACTAGTATAACTCGCAGAAGGATCAATCGATCC. $^{hc}$Y57A: FW: CCTTCTGA TAGTGCGACTAGTCACAATGAAAAGTTCAAGG, REV: CCTTGAACTTTTC ATTGTGACTAGTCGCACTATCAGAAGG. $^{lc}$S32A: FW: CCAGTCAGAACATT GGCACAGCGATCACTGGTATCAGCAAAG, REV: CTTTGCTGATACCAG TGTATCGCTGTGCCAATGTTCTGACTGG. $^{lc}$Y50A: FW: CTCCAAGGCTTAT CATAAAGGCGGCTTCTGAGTCTATCTCTGG, REV: CCAGAGATAGACTCAG AAGCCGCCTTTATGATAAGCCTTGGAG. $^{lc}$S91A: FW: CAGATTATTACTGTC AACAAGCTAATACCTGGCCGTACACGTT, REV: AACGTGTACGGCCAGG TATTAGCTTGTTGACAGTAATAATCTG. $^{lc}$W94A: FW: GTCAACAAAGTAATA CCGCGCCGTACACGTTCGGAGG, REV: CCTCCGAACGTGTACGGCGCGG TATTACTTTGTTGAC. $^{lc}$Y96A: FW: TAATACCTGGCCGGCCACGTTCGGA GGGG, REV: CCCCTCCGAACGTGGCCGGCCAGGTATTA. $^{hc}$Y101A: FW: CTGTTCAAGATCCGGCGCCGGATATTATGCTATGGAG, REV: CTCCATAG CATAATATCCGGCGCCGGATCTTGAACAG. $^{hc}$Y104A: FW: CCGGCTACGGA TATGCTGCTATGGAGTACTGGG, REV: CCCAGTACTCCATAGCAGCATA TCCGTAGCCGG), followed by subsequent expression and purification as was described for holo-POM1.

**Protein analysis.** COCS were washed twice in PBS and scraped off the PTFE membranes with PBS. Homogenization was performed with a TissueLyser LT (Qiagen) for 2 minutes at 50 Hz. A bicinchoninic acid assay (Pierce BCA Protein Assay Kit, Thermo Fisher Scientific) was used to determine protein concentrations. PrP$^{Sc}$ levels were determined through digestion of 20 μg of COCS homogenates with 25 μg mL$^{-1}$ of proteinase K (PK, Roche) at a final volume of 20 μL in PBS for 30 minutes at 37 °C. PK was deactivated by addition of sodium-dodecyl-sulfate-containing NuPAGE LDS sample buffer (Thermo Fisher Scientific) and boiling of samples at 95 °C for 5 minutes. Equal sample volumes were loaded on Nu-PAGE Bis/Tris precast gels (Life Technologies) and PrP$^C$/PrP$^{Sc}$ was detected by western blot using the monoclonal anti-PrP antibodies POM1, POM2 or POM19 at 0.4 μg mL$^{-1}$ (all holo-antibodies), as established elsewhere[8]. Further primary antibodies used for western blots in this manuscript are as follows: monomeric NeonGreen (1:1,000, 32F6, Chromotek), phospho-eIF2α (1:1,000, clone no. D9G8, Cell Signaling Technologies), eIF2α (1:1,000, clone no. D7D3, Cell Signaling Technologies), pan-actin (1:10,000, clone no. C4, Millipore), GFAP (1:1,000, clone no. D1F4Q, Cell Signaling Technologies), Iba1 (1:500, catalog no. 019–19741, Wako), NeuN (0.5 μg/ml, catalog no. ABN78, Merck Millipore) and Myc-tag (1:500, catalog no. ab9106, Abcam). After incubation of primary antibodies at 4 °C overnight, membranes were washed and detected with goat polyclonal anti-mouse (1:10,000, 115–035–062, Jackson ImmunoResearch) or goat polyclonal anti-rabbit (1:10,000, 111–035–045, Jackson ImmunoResearch) antibodies for 1 hour at room temperature. For PNGaseF digestion, 20 μg of samples was processed using a commercially available kit (New England Biolabs), and PrP$^C$ detection was performed using the monoclonal anti-PrP$^C$ antibody POM2, as described above. Western blots were quantified on native photographs (uncropped, naive images are available in the Source Data); representative western blot images in the main and supplementary figures have been processed (linear adjustment of contrast and brightness) for better visualization.

**Surface plasmon resonance (SPR).** The binding properties of the complexes between rmPrP, POM1 and pomologs were measured at 298 K on a ProteOn XPR-36 instrument (Bio-Rad) using 20 mM HEPES pH 7.2, 150 mM NaCl, 3 mM EDTA and 0.005% Tween-20 as running buffer. mPrP was immobilized on the surface of GLC sensor chips through standard amide coupling. Serial dilution of antibodies (full IgG, Fab or single-chain versions) in the nanomolar range were injected at a flow rate of 100 μL/min (contact time 6 minutes); the dissociation phase was then observed for 5 minutes. Analyte responses were corrected for unspecific binding and buffer responses by subtracting the signal of both a channel where no PrP was immobilized and a channel with no antibody was added. Curve fitting and data analysis were performed with Bio-Rad ProteOn Manager software (version 3.1.0.6).

**Statistical analyses.** All biological measurements are taken from distinct samples. Unless mentioned otherwise, the following tests were performed for statistical hypothesis testing: unpaired, two-tailed *t*-test was used for comparison between two groups, one-way ANOVA with Dunnett's multiple-comparison test was used for comparison of multiple groups with a control group, and ordinary one-way ANOVA with Šídák's multiple comparisons test was used for comparison of preselected pairs of groups. Statistical analysis and visualization were performed using Prism 8 (GraphPad). No statistical methods were used to pre-determine sample sizes, but our sample sizes are similar to those reported in previous publications[7–11]. Except for in vivo prion inoculation experiments and NeuN morphometry, data collection and analysis were not performed blind to the conditions of the experiments.

**Synchrotron radiation circular dichroism.** Secondary structure content of complexes between rmPrP and POM1 and [hc]Y57 and [hc]Y104A was analyzed with synchrotron radiation circular dichroism (SRCD) spectroscopy.

Experiments were performed using a nitrogen-flushed B23 beamline for SRCD at Diamond Light Source or ChirascanPlus CD spectropolarimeter (Applied Photophysics). With both instruments, scans were acquired at 20 °C using an integration time of 1 second and 1 nm bandwidth. Demountable cuvette cells with a pathlength of 0.00335 cm were used in the far-UV region (180–260 nm) to measure the CD of the protein concentration, varying from 10 to 102 μM protein in 10 mM NaP pH 7 and 150 mM NaCl. Mixtures were prepared to a stoichiometric molar ratio of 1:1. SRCD data were processed using CDApps[33] and OriginLab. Spectra have been normalized using an average amino acid molecular weight of 113 for secondary structure estimation from SRCD, and CD spectra were created using CDApps using the Continll algorithm[34]. For comparison of calculated and observed spectra, the full molecular weight of sample and complex were used. Measurements of free mPrP and free antibodies were taken as a reference.

**Reporting summary.** Further information on research design is available in the Nature Research Reporting Summary linked to this article.

## Data availability
All source data, for example numeric source data, uncropped western blot gels including annotations thereof, as well as unique DNA sequences, accompany this manuscript as supplements. The following publicly available data was used: Allen Mouse Brain Atlas, entries 71717640 and 227540 (https://mouse.brain-map.org); Biological Magnetic Resonance Data Bank, entry 16071 (https://bmrb.io/); RCSB Protein Data Bank, entries 1XYX and 4H88 (https://www.rcsb.org). Additionally, all unique biological materials used in the manuscript are readily available from the authors. Source data are provided with this paper.

## Code availability
New code was generated for analysis of Allen Brain Atlas data and can be found in the Supplementary Software.

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

## Acknowledgements

We would like to acknowledge M. Epskamp, T. Kottarathil, M. Carta, M. Rincon, R. Moos, J. Guo and C. Tournaire for valuable discussions and technical help, as well as T. Sonati for advising on certain experiments performed by C. Tournaire. We are grateful to G. Moro for help and discussion. Imaging was performed with equipment maintained by the Center of Microscopy and Image Analysis, University of Zurich. The viral vectors and respective plasmids were produced by the Viral Vector Facility (VVF) at the Neuroscience Center Zurich (Zentrum für Neurowissenschaften Zürich, ZNZ). We are grateful to A. P. Valente for useful discussion on protein dynamics. K. F. received unrestricted support from the Theodor und Ida Herzog-Egli-Stiftung and Ono Pharmaceuticals. R. R. R. was supported by a Career Development Award from the Stavros Niarchos Foundation. G. M. is funded by a Ramon Jenkins Research Fellowship at Sidney Sussex College. T. K. received financial support from the EPSRC, BBSRC, ERC, and the Frances and Augustus Newman Foundation. A. A. is supported by an Advanced Grant of the European Research Council (ERC, No. 250356), a Distinguished Scientist Award from the Nomis Foundation and grants from the GELU Foundation, the Swiss National Science Foundation (SNSF grant ID 179040, including a Sinergia grant), and the Swiss Initiative in Systems Biology, SystemsX.ch (PrionX, SynucleiX). L. V. gratefully acknowledges support from SNF (nos. 310030_166445, 157699), Synapsis Foundation_Alzheimer research (ARS) and Lions Club Monteceneri. We would like to thank Diamond Light Source for B23 beamtime allocation (CM-19680). M. G. H. was supported by a grant from the Thierry Latran Foundation (SOD-VIP), The Research Foundation – Flanders (FWO, grant 1513616 N), European Research Council (ERC) Proof of Concept Grant 713755 – AD-VIP) and the European Commission (H2020-WIDESPREAD-2018–2020–6; NCBio; 951923).

## Author contributions

A. A. and L. V. conceived and supervised the project, acquired funds and provided experimental resources. K. F. conceptualized, performed and analyzed slice culture experiments (except those depicted in Figure 7) and in vitro and AAV therapy experiments. M. B. cloned, produced and characterized all scFv antibodies used in this manuscript. NMR experiments were performed by M. B., L. S., S. J. and O. Z.. M. B. conceptualized, performed and analyzed MDS and SPR experiments with support from M. P., F. M. and L. S.. A. S. conceptualized, performed and analyzed all experiments related to phage-display technology and, together with A. H., cloned and performed pilot AAV pomolog experiments. R. R. R. conceptualized, performed and analyzed all MEMRI experiments. S. B. performed and analyzed experiments depicted in Figures 2d and 6h–j under the supervision of K. F. CD Synchrotron experiments were conceptualized, performed and analyzed by R. H. and G. S. M. M. and M. G. H. conceived and produced in vivo AAV pomolog therapies. S. H. and C. Z. supervised and conceptualized experiments carried out by K. F. P. S. performed in vivo prion inoculations, AAV transductions and necropsies. M. L. designed and cloned non-therapeutic AAV and performed slice culture experiments under the supervision of K. F.. G. M. and T. K. conceptualized and analyzed FRET pomolog binding data. A. L. conceptualized, performed and analyzed in vitro vacuolation experiments. K. F., M. B., M. P., L. S., L. V. and A. A. wrote the original and revised manuscript. All authors performed critical review of both the original and revised version of the manuscript.

## Competing interests
The authors declare no competing interests. The funders had no role in study design, data collection and analysis, decision to publish or preparation of the manuscript.

## Additional information

**Extended data** is available for this paper at https://doi.org/10.1038/s41594-022-00814-7.

**Correspondence and requests for materials** should be addressed to Luca Varani or Adriano Aguzzi.

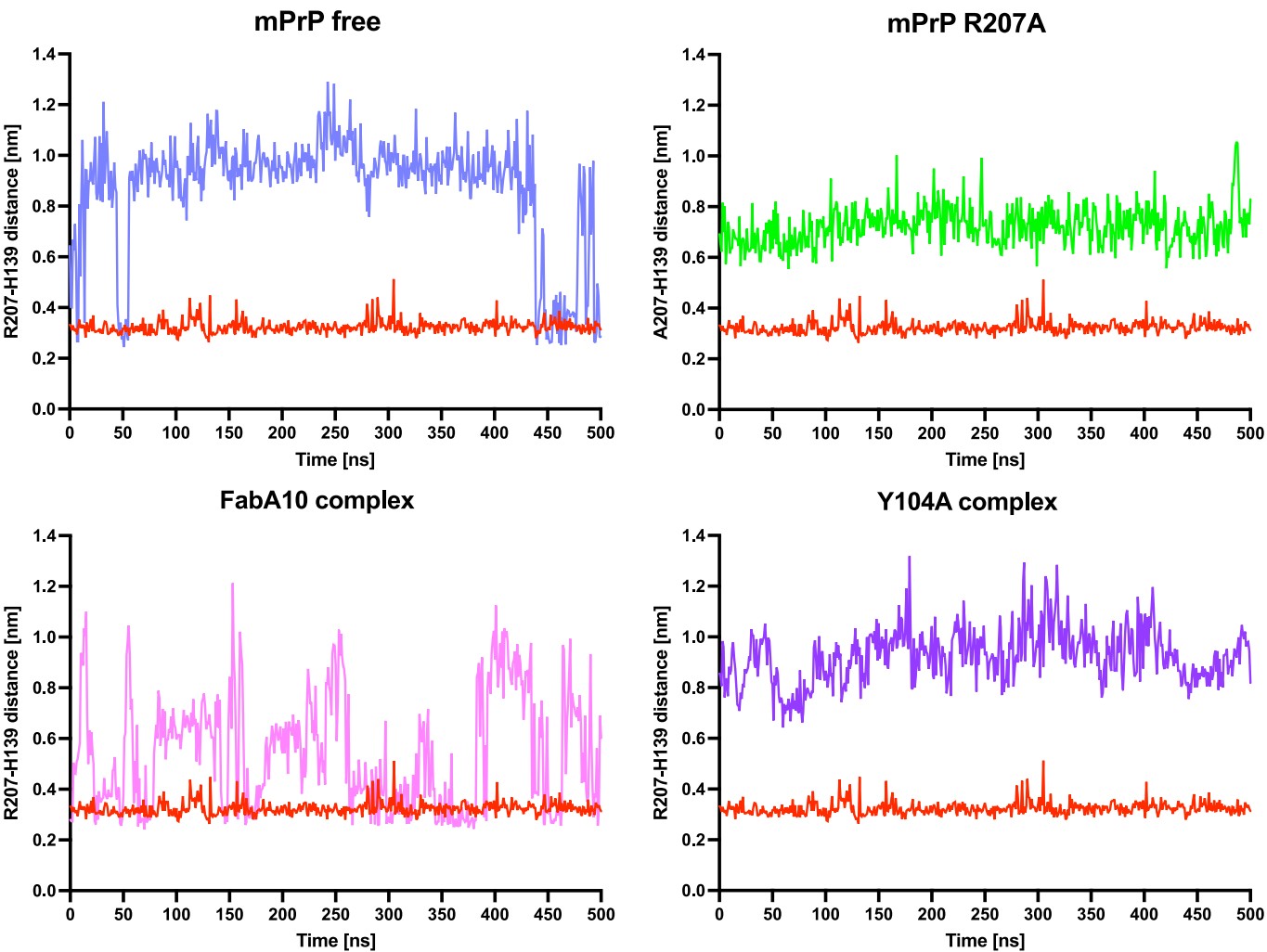

**Extended Data Fig. 1 | Distances between the R207 and H139 residues in MD simulations of PrP-antibody complexes.** The simulation of the PrP-POM1 complex (red) in reproduced in all charts to facilitate comparisons. When PrP is bound to POM1, the H-bond between R207 and H139 (termed H-latch) is always present, with distance between the centroid of their sidechains around 0.3 nm. Greater distances indicate loss of hydrogen interactions and consequently absence of the H-latch. The complex of PrP with the [hc]Y104A pomolog never shows formation of the H-latch, whereas FabA10 shows intermediate values. Simulations were run three times, but only representative traces are shown; aggregated analyses are shown in Fig. 1c.

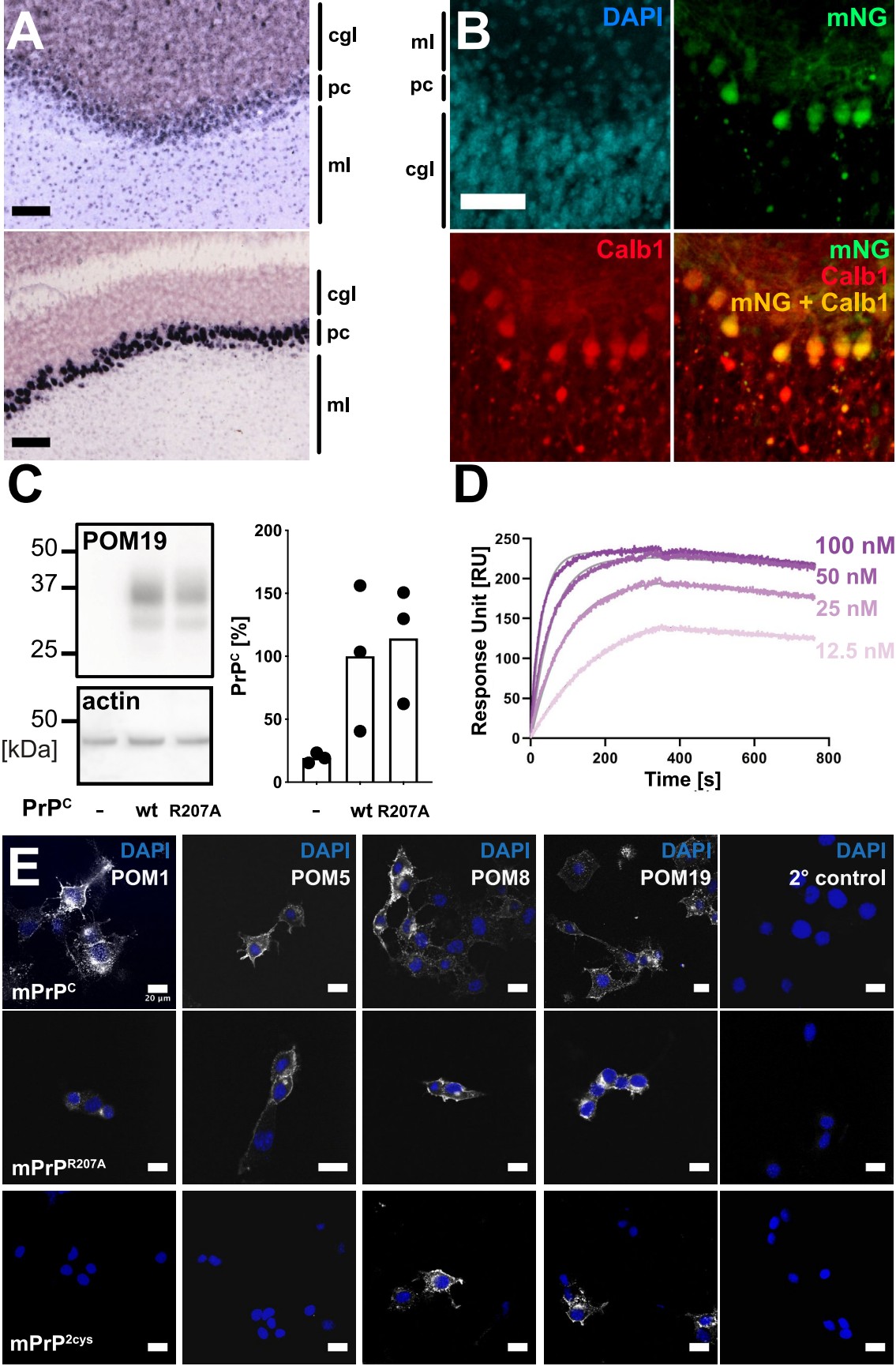

**Extended Data Fig. 2 | See next page for caption.**

**Extended Data Fig. 2 | Robust expression and conformation of the PrPR207A point mutant. (a)** Representative images of expression levels of Synapsin 1 (Syn1, *upper*) and Calbindin 1 (Calb1, *lower*) show predominant (Syn1) or almost exclusive (Calb1) expression in Purkinje cells (pc) in the cerebellar cortex. Image credit: Allen Institute. Scale bar = 100 μm. **(b)** Fluorescent micrographs of $Prnp^{ZH3/ZH3}$ COCS transduced with the AAV outlined in panel (A) show mNeonGreen expression predominantly in calbindin 1-expressing Purkinje cells. Scale bar = 50 μm. cgl = cerebellar internal granular layer, pc = Purkinje cell layer, ml = molecular layer. These findings were repeated in three independent experiments. **(c)** *Left panel*: Stably transfected CAD5-mPrP$^C$ and CAD5-mPrP$^{R207A}$ cells show similar PrP$^C$ expression levels. epresentative PrP$^C$ levels of one cell culture passage are shown. *Right panel*: POM19 immunoreactivity is divided by actin immunoreactivity, values are given as percentages of PrP$^C$. One datapoint corresponds to one passage of CAD5 cells. **(d)** Surface plasmon resonance (SPR) traces showing binding of POM1 to recombinant mPrP$^{R207A}$ (rmPrP$^{R207A}$, $k_a$=3.8E$^{+05}$ 1/Ms, $k_d$=1.8E$^{-04}$ 1/s, $K_D$=4.7E$^{-10}$ M; for comparison binding to recombinant wild-type murine PrP showed $k_a$=3.6E$^{+05}$ 1/Ms; $k_d$=9.1E$^{-05}$ 1/s; $K_D$=2.5E$^{-10}$ M). **(e)** Immunohistochemistry of CAD5 $Prnp^{-/-}$ cells stably transfected with pcDNA3.1 vector expressing wild-type murine PrP$^C$ (mPrP$^C$), mPrP$^{R207A}$ and mPrP$^{2cys}$. Monoclonal anti-PrP$^C$ antibodies targeting distinct conformational epitopes on the globular domain of PrP$^C$ were incubated to assess conformational changes in mPrP$^{R207A}$ (POM1: α1-α3, POM5: β2-α2, POM8: α1-α2, POM19: β1-α3). Except for diminished staining of POM1 in mPrP$^{R207A}$, we observed robust detection of mPrP$^{R207A}$ by POM5, POM8 and POM19 and mPrP$^{2cys}$ by POM8 and POM19. Parts of this experiment, for example POM1 and POM19, were repeated twice. Scale bar = 20 μm.

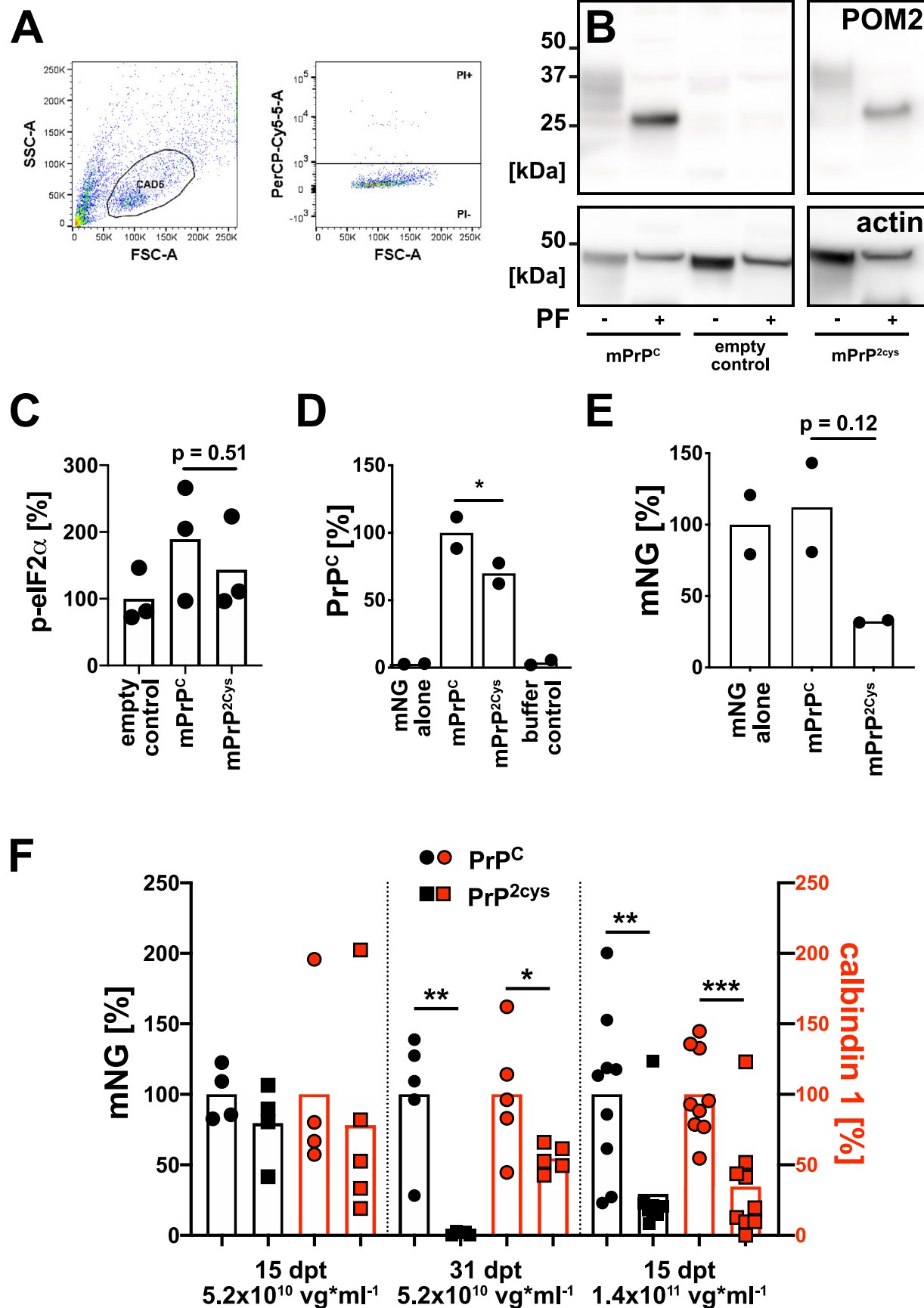

**Extended Data Fig. 3 | See next page for caption.**

**Extended Data Fig. 3 | Expression of the H-latch mimic R207C-I138C in organotypic cultured slices leads to dose-dependent neurotoxicity. (a)** Flow cytometry gating strategy of PI positive CAD5 cells. **(b)** PNGase-F digestion of cell lysates induced a shift in both murine wild-type PrP$^C$ and mPrP$^{2cys}$, indicating that both moieties had undergone N-linked glycosylation to a similar extent. Non-adjacent lanes were merged from the same gel. **(c)** CAD5 *Prnp*$^{-/-}$ cells expressing mPrP$^{2cys}$ did not show an upregulation of the unfolded protein response, suggesting that mPrP$^{2cys}$ did not undergo pathological degradation. Values are given as percentage of empty control vector (p-eIF2α / eIF2α / actin). One datapoint per group corresponds to a different cell culture passage. Two-sided, unpaired t-test. **(d)** A POM2/POM3 sandwich ELISA of COCS transduced with empty control, mPrP$^C$, mPrP$^{2cys}$ and buffer control shows robust mPrP$^{2cys}$ expression in transduced COCS, albeit significantly less than wild-type mPrP$^C$. Slices were harvested at 28 days post-transduction. One datapoint corresponds to an independent, biological replicate of 6–9 pooled slices. Ordinary, one-way Anova with Šídák's multiple comparisons test, *: adjusted p-value = 0.039 **(e)** Reduced levels of mNG in *Prnp*$^{-/-}$ (ZH3) COCS expressing mPrP$^{2cys}$. mNG immunoreactivity values are divided by actin immunoreactivity and expressed as percentages of empty control. Slices were harvested at 28 days post-transduction. One datapoint corresponds to an independent, biological replicate of 6–9 pooled slices. Ordinary, one-way Anova with Šídák's multiple comparisons test. Raw, uncropped blots can be found in the Source Data supplement. **(f)** Quantification of mNG and Calb1 fluorescence intensity from experiments shown in Fig. 3d-f. One datapoint corresponds to a biological replicate, e.g. one organotypic cultured slice. Unpaired, two-tailed t-test without adjustment for multiple testing. P-values are as follows: 31 dpt, 5.2x10$^{10}$ vg*ml$^{-1}$, mNG: 0.001; 31 dpt, 5.2x10$^{10}$ vg*ml$^{-1}$, Calb1: 0.0496; 15 dpt, 1.4x10$^{11}$ vg*ml$^{-1}$, mNG: 0.0065; 15 dpt, 1.4x10$^{11}$ vg*ml$^{-1}$, Calb1: 0.001. ***: p ≤ 0.001, **: p < 0.01, * p < 0.05.

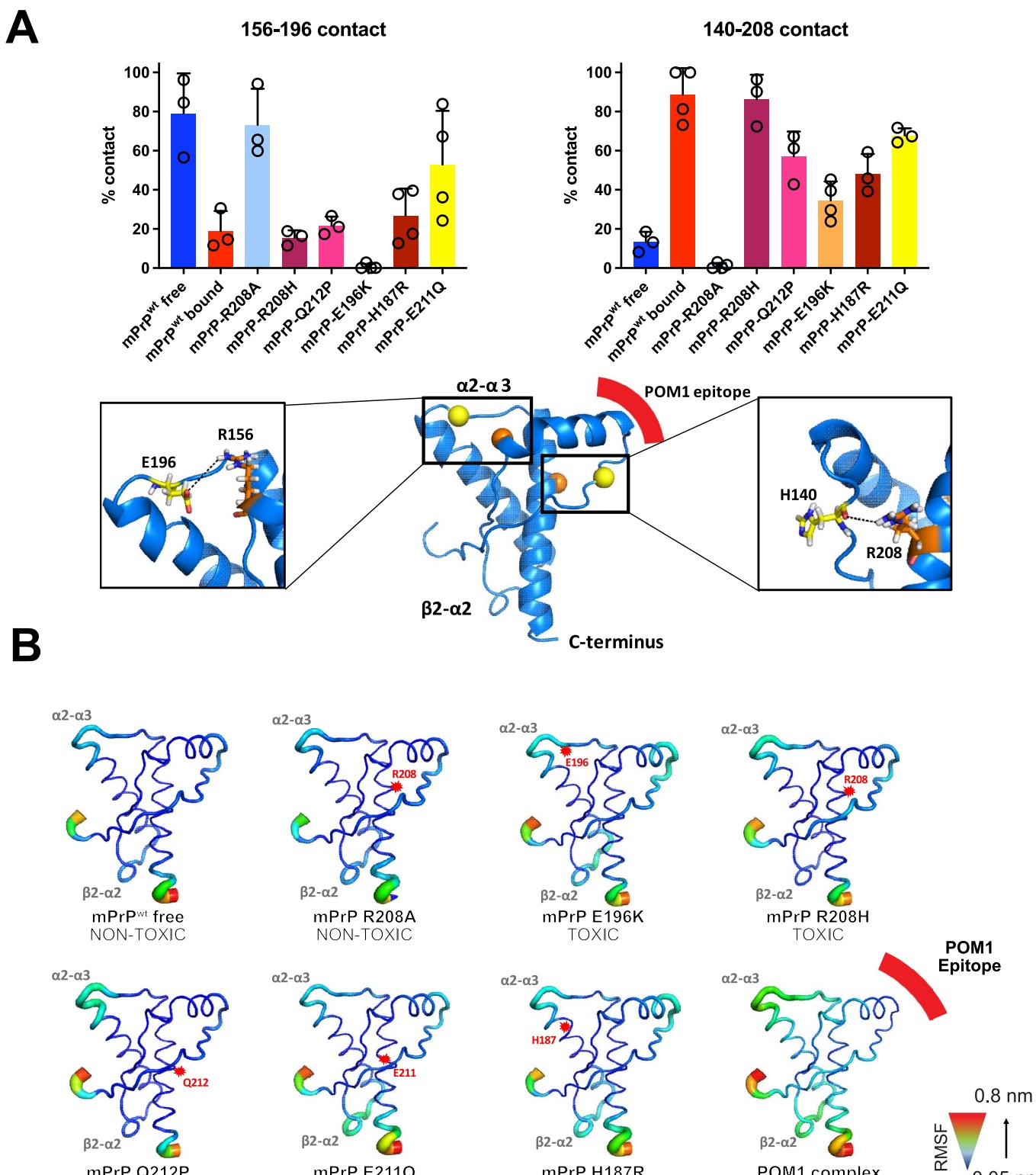

**Extended Data Fig. 4 | Molecular dynamics simulations show overlapping structural changes of POM1-PrP^C complex and pathogenic *PRNP* mutations.**
Extended Data Figure 4. **(a)** MD simulations of POM1 binding and pathogenic *PRNP* mutations causing genetic prion disease show the R156-E196 interaction is abolished and induction of the H140-R208 H-latch is established. Each datapoint represents one independent simulation, values are given as mean ± standard deviation. **(b)** In agreement with this view, POM1 and human, hereditary PrP mutations responsible for fatal prion diseases favor altered flexibility in the α2-α3 and β2-α2 loop.

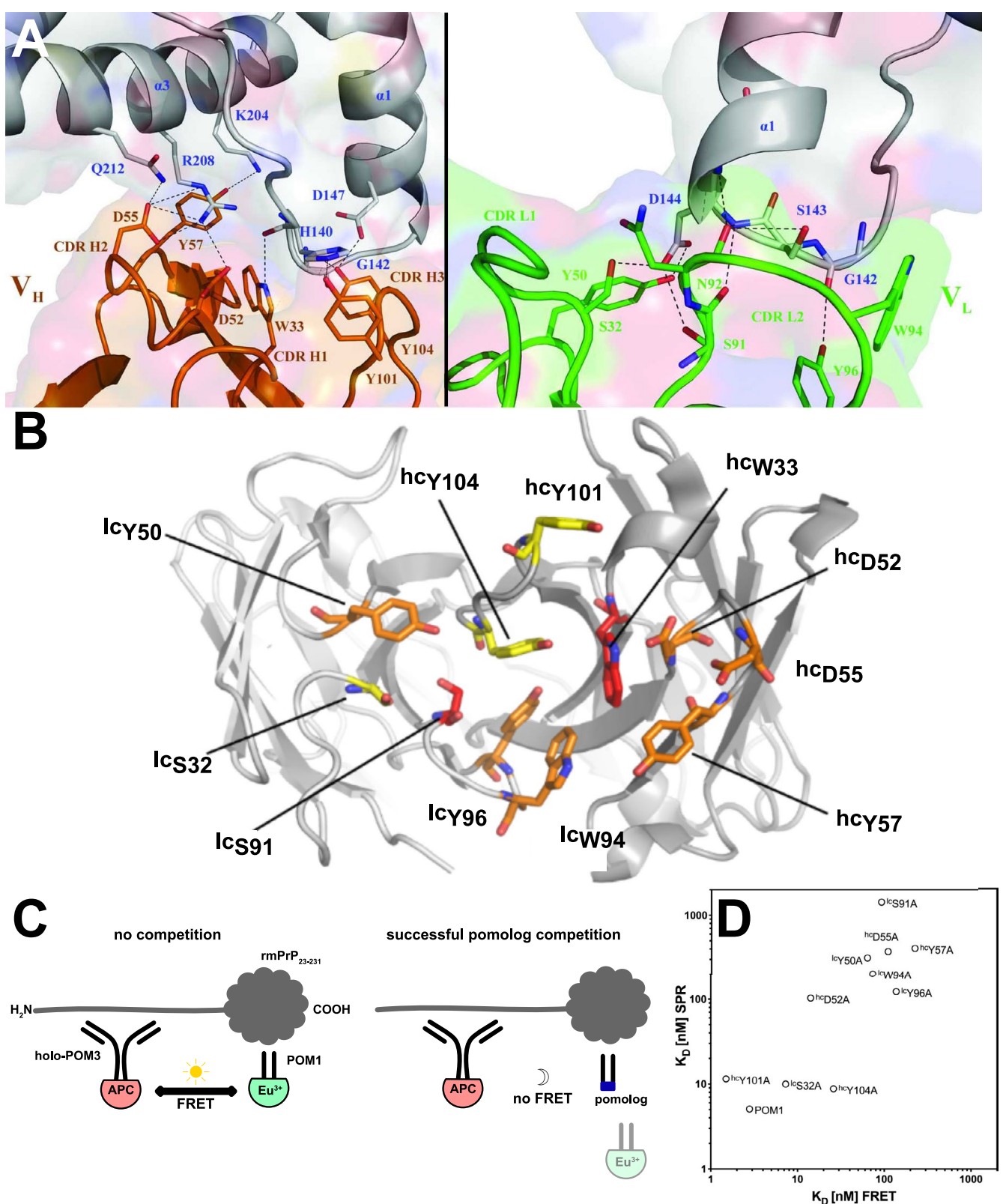

**Extended Data Fig. 5 | See next page for caption.**

**Extended Data Fig. 5 | Scanning alanine mutagenesis of the POM1 paratope.** Extended Data Figure 5. **(a)** Intermolecular contacts between human PrP$^C_{120-230}$ and POM1 Fab variable heavy chain (magenta, *left panel*), and POM1 Fab variable light chain (green, *right panel*) as determined by Baral et al., 2012[8]. Reproduced with permission of the International Union of Crystallography from doi:10.1107/S0907444912037328. **(b)** Schematic representation of a single-chain fragment of wild-type POM. The mutated residues are indicated as stick on the cartoon structure of POM1, color coded as in Supplementary Table 1b. The CDR loops are shown from the perspective of the antigen. **(c)** Scheme of competition FRET assay to assess the K$_D$ of various pomologs. In the absence of competing antibody, FRET occurs due to proximity of allophycocyanin (APC)-labeled holo-POM3 and europium (Eu3+)-labeled POM1 (*left panel*). Because of liquid-phase competition, addition of unlabeled pomologs leads to a decrease in FRET signal (*right panel*). The calculation of binding constants from FRET is detailed in the methods section. **(d)** The binding constants measured by SPR and by FRET were in good agreement, Spearman r = 0.77, p=0.0074, 95% CI 0.30–0.94) with the exception of $^{hc}$W33A, whose binding on SPR was too weak to be precisely measured.

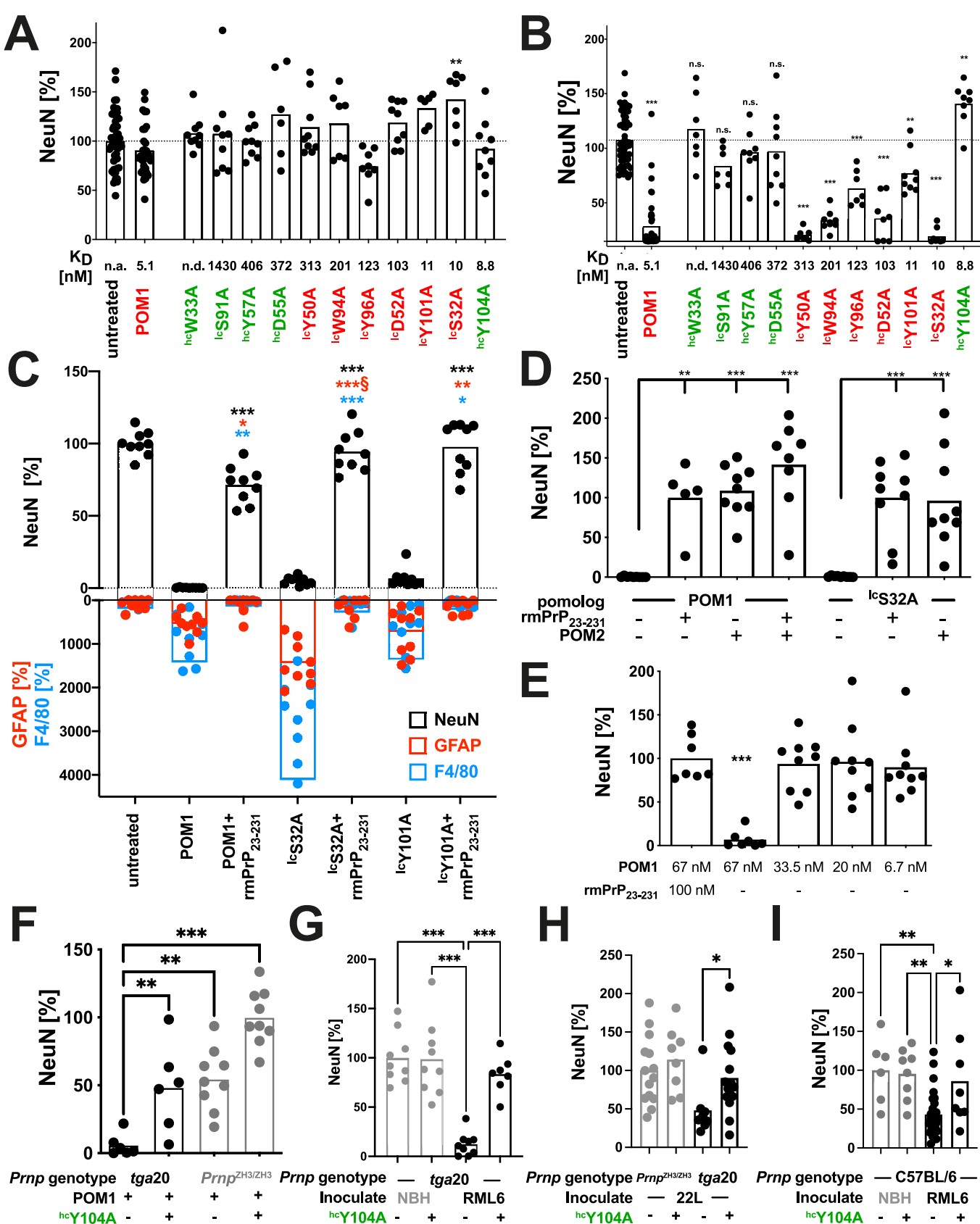

**Extended Data Fig. 6 | See next page for caption.**

**Extended Data Fig. 6 | Pomolog $^{hc}$Y104A acts as dominant negative suppressor of prion toxicity.** Extended Data Figure 6. **(a)** Treatment of $Prnp^{ZH1/ZH1}$ COCS shows toxicity of POM1 and toxic pomologs to be dependent on $PrP^{C}$, see also Supplementary Fig. 1a. **: p=0.003, ordinary one-way Anova with Dunnett's multiple comparisons test. Innocuous pomologs are highlighted in green, POM1 and toxic pomologs are highlighted in red. **(b)** Morphometric quantification of $Prnp$-overexpressing tga20 COCS treated with pomologs, see also Fig. 4. Color coding according to panel (A). 100%=untreated COCS, comparison of untreated versus treated groups. N.s.: not significant, ***: p < 0.0001, ** $^{hc}$Y101A: p=0.0035, ** $^{hc}$Y104A: p=0.0019, ordinary one-way Anova with Dunnett's multiple comparisons test. **(c)** Morphometric quantification of fluorescence intensity from images depicted in Supplementary Fig. 1b. §: 1 outlier was excluded (y=2046.3%, p < 0.05, extreme studentized deviate method). Values = % of untreated control. Pairwise comparison in the presence or absence of $rmPrP_{23-231}$. ***: p < 0.0001, * GFAP-POM1: p=0.0148, ** GFAP-$^{lc}$Y101A: p=0.0009, ** F4/80-POM1: p=0.0005, * F4/80-$^{lc}$Y101A: p=0.0261, ordinary one-way Anova with Šídák's multiple comparisons test. **(d)** Toxicity of high-affinity pomolog $^{lc}$S32A ablated by POM2. 100%=POM1 + $rmPrP_{23-231}$ (*bars 1–4*) or 100%=$^{lc}$S32 + $rmPrP_{23-231}$ (*bars 5–7*). ** p=0.0003, *** p < 0.0001, ordinary one-way Anova with Šídák's multiple comparisons test. **(e)** Titration of minimal toxic dosage of POM1 in tga20 COCS. 100%=POM1 + $rmPrP_{23-231}$. ***: p < 0.0001, ordinary one-way Anova with Dunnett's multiple comparisons test. **(f)** $^{hc}$Y104A prevented POM1-induced toxicity. 100%=$Prnp^{0/0}$ COCS treated with POM1 + $^{hc}$Y104A. ** p=0.0078, ***(left): p=0.0009, ***(right): p < 0.0001, ordinary one-way Anova with Dunnett's multiple comparisons test. **(g)** Quantification of Fig. 4b. 100%=untreated+NBH. ***: p < 0.0001, ordinary one-way Anova with Dunnett's multiple comparisons test. **(h)** Quantification of Fig. 4c. 100% = $Prnp^{ZH3/ZH3}$ + 22 L. *: p=0.032, ordinary one-way Anova with Šídák's multiple comparisons test. **(i)** Quantification of Fig. 4d. 100%=untreated +NBH. *: p=0.0203, **(left): p=0.0036, **(right): p=0.005, ordinary one-way Anova with Dunnett's multiple comparisons test. All graphs: one datapoint corresponds to one biological replicate.

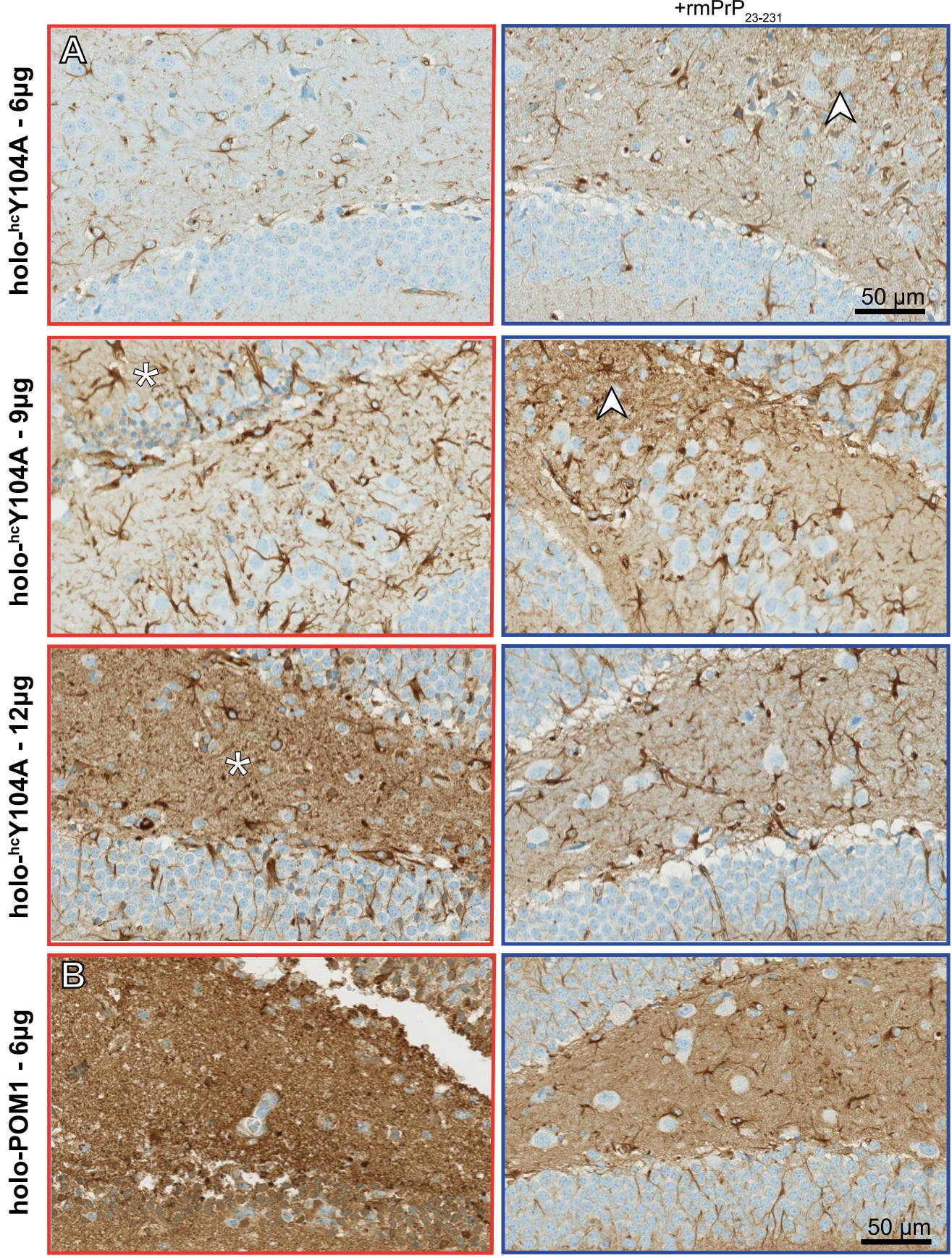

**Extended Data Fig. 7 | See next page for caption.**

**Extended Data Fig. 7 | Dose-dependent gliosis of $^{hc}$Y104A is also conspicuous around needle tracts.** Extended Data Figure 7. (**a**) Photomicrographs of glial fibrillary acid protein (GFAP) immunohistochemistry on consecutive sections depicted in Fig. 6c. *Left column*: holo-$^{hc}$Y104A injections (6, 9 and 12 μg). *Right column*: holo-$^{hc}$Y104A + rmPrP$_{23-231}$. GFAP immunoreaction was increased in areas of neuronal damage (*white asterisks*) and around needle tracts (*white arrowheads*). (**b**) Micrographs demonstrating an intensive GFAP immunoreaction in areas with extensive holo-POM1 (6 μg)- induced neurotoxicity. *Left panel*: POM1 injection (6 μg). *Right panel*: holo-$^{hc}$Y104A + rmPrP$_{23-231}$. Sections are consecutive to those shown in Fig. 6f.

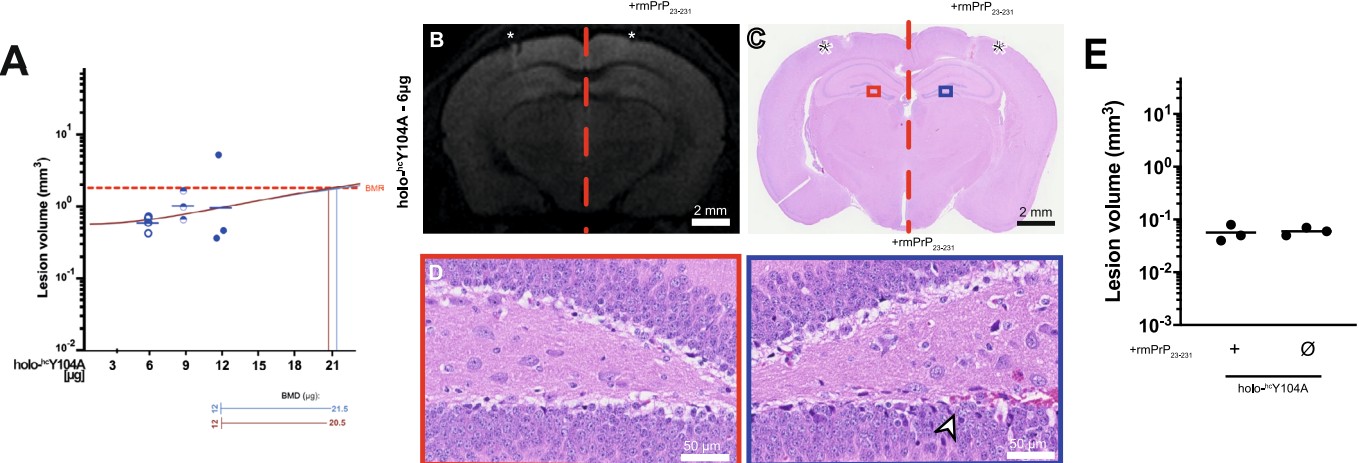

**Extended Data Fig. 8 | Assessing hcY104A dose-dependent toxicity.** Extended Data Figure 8. **(a)** A hypothetical benchmark dose analysis was performed using $\log_{10}$-transformed lesion volumes corresponding to different amounts of holo-hcY104A (data from Fig. 6g). BMR: Benchmark response (0.15 mm³, dashed red line). The benchmark dose (BMD) is defined as the dose at the BMR. The vertical lines indicate the BMD values corresponding to the different dose response values (blue: 21.5 µg, brown line: 20.5 µg). The upper limit of the safe dose is provided by the lower 95% confidence interval of the BMD (horizontal lines below the graph: blue: 12 µg, brown: 12 µg). One datapoint corresponds to one independent animal. **(b)** Representative DWI images taken 24 h after stereotactic injection of 6 µg holo-hcY104A into male tga20 mice (*left half of the image*, injected into CA3). Contralateral side: 6 µg holo-hcY104A pre-incubated with an equimolar amount of rmPrP$_{23-230}$. *White asterisks*: needle tract. **(c)** Photomicrograph of HE-stained sections from mouse brain shown in panel B. *Asterisks*: needle tract. Rectangles correspond to regions magnified in panel C. **(d)** Higher magnification of the end-plate of the hippocampus. *Left panel*: holo-hcY104A. *Right panel*: holo-hcY104A preincubated with rmPrP$_{23-230}$. *Arrow*: needle tract. **(e)** Quantification of lesion volumes after injection of holo-hcY104A in contrast to control injection into tga20 mice (N = 3). One datapoint corresponds to one independent animal.

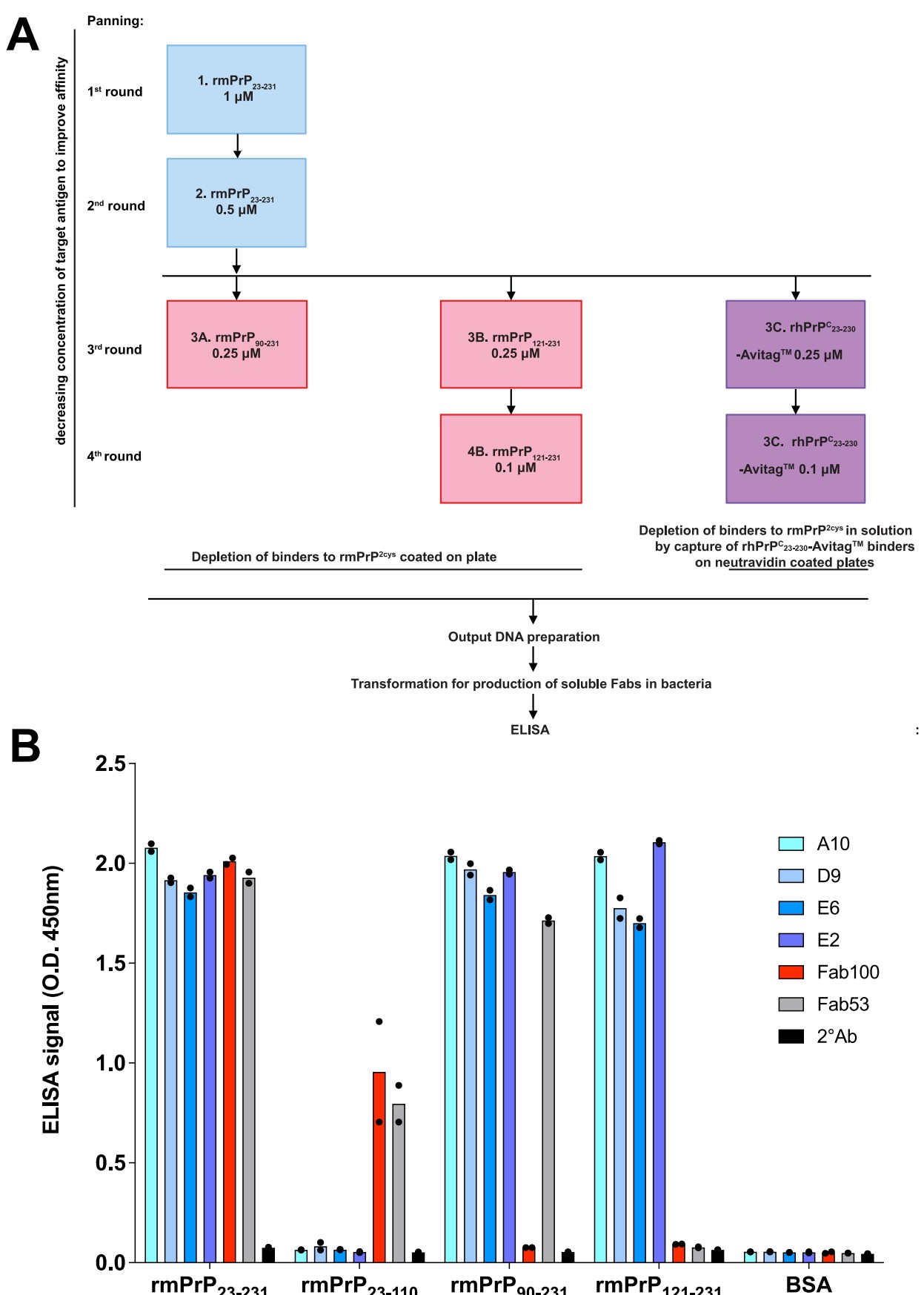

**Extended Data Fig. 9 | See next page for caption.**

**Extended Data Fig. 9 | Generation and validation of a synthetic human Fab phage library.** Extended Data Figure 9. **(a)** A synthetic human Fab phage library was used for panning. For each panning round, the targeted antigens are reported with the respective concentration. Full-length recombinant murine PrP$_{23-231}$ (rmPrP$_{23-231}$; light blue boxes) was used as a target for the first and the second round of phage panning. At the third and fourth round, phages were depleted of the binders to rmPrP$^{2cys}$ and selected for binding to either rmPrP$_{90-231}$ or rmPrP$_{121-231}$ (recombinant murine PrP fragments lacking the N-terminal flexible tail; light red boxes) or to recombinant human PrP$_{23-230}$-AviTag™ (rhPrP$_{23-230}$-AviTag™, purple boxes). In rmPrP$_{90-231}$ or rmPrP$_{121-231}$ panning, Fab-displayed Fab were depleted of binders to rmPrP$^{2cys}$ coated on plates. In rhPrP$_{23-230}$-AviTag™ panning, depletion of binders to rmPrP$^{2cys}$ in solution was achieved by capturing Fabs binding to rhPrP$_{23-230}$-AviTag™ on neutravidin coated wells. Polyclonal DNA preparation from the selected phages at the third round (rmPrP$_{90-231}$) and fourth round (rmPrP$_{121-231}$ and rhPrP$_{23-230}$-AviTag™) was used for transformation in bacteria and the screening of single clones by ELISA. **(b)** ELISA (OD at 450 nm) comparing the reactivity of phage-derived anti-PrP Fabs to full-length rmPrP$_{23-231}$, FT fragment rmPrP$_{23-110}$ and GD fragments rmPrP$_{90-231}$ and rmPrP$_{121-231}$. Anti-PrP Fab100 and Fab53 bind within the FT of PrP - the octapeptide repeat region (OR, amino acid 51–90) and the charged cluster 2 (CC2, amino acid 93–100), respectively. FabA10, FabD9, FabE6 and FabE2 bind within the GD. Error bars = standard error of the mean. One datapoint corresponds to a technical replicate in a multi-well plate.

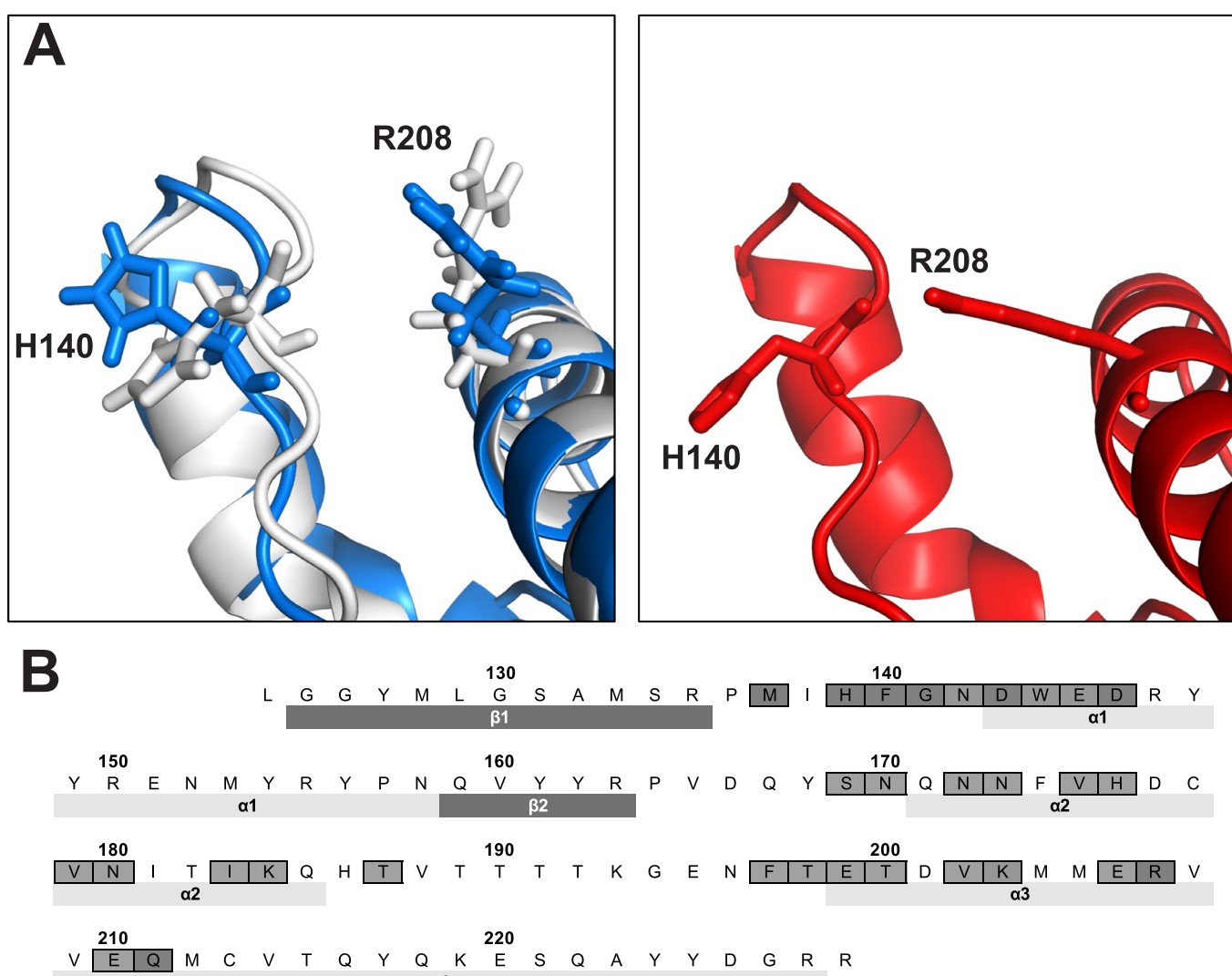

**Extended Data Fig. 10 | FabA10 ameliorates the H-latch but shares its paratope with POM1.** Extended Data Figure 10. **(a)** The R208-H140 interaction is present in POM1-bound PrP (right, *red*) but not in free PrP (left, *white*) or in its complex with FabA10 (left, *blue*). The final state of MD simulations starting from a POM1-bound conformation, with R208-H140 interaction present, is shown for FabA10. **(b)** Overlap (green) of FabA10 (blue) and POM1 (red) epitopes on murine PrP$^C$-GD. Coloring according to Fig. 7d, e.

Luca Varani
Adriano Aguzzi

# Reporting Summary

## Statistics

For all statistical analyses, confirm that the following items are present in the figure legend, table legend, main text, or Methods section.

| n/a | Confirmed | |
|---|---|---|
| ☐ | ☒ | The exact sample size (*n*) for each experimental group/condition, given as a discrete number and unit of measurement |
| ☐ | ☒ | A statement on whether measurements were taken from distinct samples or whether the same sample was measured repeatedly |
| ☐ | ☒ | The statistical test(s) used AND whether they are one- or two-sided <br> *Only common tests should be described solely by name; describe more complex techniques in the Methods section.* |
| ☐ | ☒ | A description of all covariates tested |
| ☐ | ☒ | A description of any assumptions or corrections, such as tests of normality and adjustment for multiple comparisons |
| ☐ | ☒ | A full description of the statistical parameters including central tendency (e.g. means) or other basic estimates (e.g. regression coefficient) AND variation (e.g. standard deviation) or associated estimates of uncertainty (e.g. confidence intervals) |
| ☐ | ☒ | For null hypothesis testing, the test statistic (e.g. *F*, *t*, *r*) with confidence intervals, effect sizes, degrees of freedom and *P* value noted <br> *Give P values as exact values whenever suitable.* |
| ☒ | ☐ | For Bayesian analysis, information on the choice of priors and Markov chain Monte Carlo settings |
| ☒ | ☐ | For hierarchical and complex designs, identification of the appropriate level for tests and full reporting of outcomes |
| ☐ | ☒ | Estimates of effect sizes (e.g. Cohen's *d*, Pearson's *r*), indicating how they were calculated |

*Our web collection on statistics for biologists contains articles on many of the points above.*

## Software and code

Policy information about availability of computer code

| Data collection | We used RStudio (1.1.383) with R (3.4.2), CDApps (4.0), FlowJo (10) |
|---|---|
| Data analysis | Custom code was used to analyze Allen Mouse Brain Atlas data, which can be found in the supplementary data. |

For manuscripts utilizing custom algorithms or software that are central to the research but not yet described in published literature, software must be made available to editors and reviewers. We strongly encourage code deposition in a community repository (e.g. GitHub). See the Nature Portfolio guidelines for submitting code & software for further information.

## Data

Policy information about availability of data

All manuscripts must include a data availability statement. This statement should provide the following information, where applicable:
- Accession codes, unique identifiers, or web links for publicly available datasets
- A description of any restrictions on data availability
- For clinical datasets or third party data, please ensure that the statement adheres to our policy

All source data, e.g. numeric source data, uncropped western blot gels including annotation thereofs as well as unique DNA sequences, is available in the manuscript or the supplemen-tary materials. The following, publicly available data was used: Allen Mouse Brain Atlas, entries 71717640 and 227540 (https://mouse.brain-map.org); Biological Magnetic Resonance Data Bank, entry 16071 (https://bmrb.io/); RCSB Protein Data Bank, entries 1XYX and 4H88 (https://www.rcsb.org). Additionally, all unique biological materials used in the manuscript are readily available from the authors.

# Human research participants

Policy information about studies involving human research participants and Sex and Gender in Research.

| | |
|---|---|
| Reporting on sex and gender | *Use the terms sex (biological attribute) and gender (shaped by social and cultural circumstances) carefully in order to avoid confusing both terms. Indicate if findings apply to only one sex or gender; describe whether sex and gender were considered in study design whether sex and/or gender was determined based on self-reporting or assigned and methods used. Provide in the source data disaggregated sex and gender data where this information has been collected, and consent has been obtained for sharing of individual-level data; provide overall numbers in this Reporting Summary. Please state if this information has not been collected. Report sex- and gender-based analyses where performed, justify reasons for lack of sex- and gender-based analysis.* |
| Population characteristics | *Describe the covariate-relevant population characteristics of the human research participants (e.g. age, genotypic information, past and current diagnosis and treatment categories). If you filled out the behavioural & social sciences study design questions and have nothing to add here, write "See above."* |
| Recruitment | *Describe how participants were recruited. Outline any potential self-selection bias or other biases that may be present and how these are likely to impact results.* |
| Ethics oversight | *Identify the organization(s) that approved the study protocol.* |

Note that full information on the approval of the study protocol must also be provided in the manuscript.

# Field-specific reporting

Please select the one below that is the best fit for your research. If you are not sure, read the appropriate sections before making your selection.

☒ Life sciences ☐ Behavioural & social sciences ☐ Ecological, evolutionary & environmental sciences

For a reference copy of the document with all sections, see nature.com/documents/nr-reporting-summary-flat.pdf

# Life sciences study design

All studies must disclose on these points even when the disclosure is negative.

| | |
|---|---|
| Sample size | No statistical methods were used to predetermine sample size, moreover, sample size was chosen based on previously performed experiments, as published before (PMID 26821311, 25710374, 23903654, 23133383) |
| Data exclusions | ED Fig 6C: immunohistochemical analysis of slice cultures is known to show large variability (Falsig et al., PLOS Pathogens 2012, PMID 23133383) , here, one extreme outlier was excluded from the analysis (y=2046.3%, p<0.05, extreme studentized deviate method). |
| Replication | Biological replicates from in vitro experiments correspond to different culture passages. Biological replicates from organtypic slice culture correspond to slices from different animals. Experiments depicted in Figures 2D, 3D-F, 4A, 4B, 4D+F, 7A, ED 3E, ED 6A,B,G,I were successfully replicated in two independent runs. Experiments depicted in Figures 1C,D,F, 2B+C, 3B+C, 6D+E, ED1, ED4, ED5D were successfully replicated in three or more independent runs. Experiments depicted in the following images contain multiple, independent, biological replicates, but were only performed once: 2E, 2F, 3A, 4C, 4E, 4G, 5A, ED 3B, ED 6B-F, ED 6H, ED7B+C. Experiments depicted in ED 3C+D were performed once with two independent, biological replicates. POM1 and POM19 of experiment from Fig ED 2D were replicated successfully, we did not attempt replication of POM5 and POM8, because it was later orthogonal verified in MDS and NMR. Due to disproportionate effort and in agreement with standard practice, intracerebral/-venous injections and in vivo imaging of animals (depicted in Fig. 6, ED 7+8, although they were performed with several independent, biological replicates), NMR and SRCD analyses (Fig. 3, 5, 7D+E) as well as antibody phage display (Fig. ED 9) were only performed once. |
| Randomization | Animals were randomly assigned to treatment groups. Brain slices were pooled from different animals and randomly selected for treatment. Similarly, allocation of cells in in vitro experiments was random. Sections used for imaging were selected randomly and were analyzed equally with no sub-sampling thus omitting a need for randomization. |
| Blinding | NeuN morphometry was performed blindly. Prion disease symptoms of AAV-treated mice were assessed by an animal caretaker blinded to the treatment (P.S.). In the case of other experiments, blinding was not attempted, in parts because testing conditions were evident from the experimental data. However, quantifications were performed using computational pipelines applied equally to all conditions and replicates for a given experiment. |

# Reporting for specific materials, systems and methods

We require information from authors about some types of materials, experimental systems and methods used in many studies. Here, indicate whether each material, system or method listed is relevant to your study. If you are not sure if a list item applies to your research, read the appropriate section before selecting a response.

## Materials & experimental systems

| n/a | Involved in the study |
|-----|----------------------|
| ☐ | ☒ Antibodies |
| ☐ | ☒ Eukaryotic cell lines |
| ☒ | ☐ Palaeontology and archaeology |
| ☐ | ☒ Animals and other organisms |
| ☒ | ☐ Clinical data |
| ☒ | ☐ Dual use research of concern |

## Methods

| n/a | Involved in the study |
|-----|----------------------|
| ☒ | ☐ ChIP-seq |
| ☐ | ☒ Flow cytometry |
| ☐ | ☒ MRI-based neuroimaging |

## Antibodies

**Antibodies used**

Please specify the original source (commercial or other. If they are commercially available, please provide information on their supplier name, catalog number, clone name and lot number, as applicable) for all the other POM whole IgG antibodies, here in the reporting summary.

Please ensure all the antibodies referenced in the manuscript are listed here in the reporting summary (with supplier name, catalog number, clone name and lot number, as applicable). For example: Alexa Fluor 594 Rabbit Anti-Goat (IgG) secondary antibody

Please provide the catalog number, and lot number, as applicable, for all the following antibodies listed, here in the reporting summary. For example: phospho-eIF2α (Cell Signaling Technologies), etc.

POM1 and POM2 whole IgG antibodies are available via Merck (POM1 MABN2285, POM2 MABN2298).
Other POM whole IgG antibodies are originially derived by immunization of Prnp knock-out mice and subsequent monoclonal expansion of hybridomas as laid out in Polymenidou et al., PLOS ONE 2008, PMID 19060956. Generation of Fab and scFv fragments was performed using pepsin digestion of holo-IgGs and periplasmic expression, respectively, as reported extensively in Sonati et al., Nature 2013, PMID 23903654. Generation procedures of POM1 holo-IgG and scFv mutants by eukaroyotic and periplasmic expression, respectively, is listed in the materials and methods section of the present manuscript, the antibodies can be obtained through the authors.

Antibody sequences are deposited on NCBI (POM1 heavy chain 4DGI_H, POM1 light chain 4DGI_L, POM2 heavy chain 4J8R_D, POM2 light chain  4J8R_C) and in Polymenidou et al., PLOS One 2008, or are described in Materials and Methods (POM1 mutants) or can be obtained from the authors (Fab fragments displayed in Fig 7A).

Other antibodies used are as follows, given as Name, Catalog #, Company
anti-Fd, PC075, Binding Site
anti-Myc-tag , AB9106, Abcam
anti-monomeric NeonGreen, 32F6-100, Chromotek
anti-phospho-eIF2α, 3398, Cell Signaling Technologies
anti-eIF2α, 5324, Cell Signaling Technologies
anti-Actin Antibody, MAB1501R, Millipore
anti-GFAP, Z0334, DAKO/Agilent
anti-F4/80, MCA497G, Serotec/BioRad
anti-human F(ab')2-alkaline phosphatase conjugated antibody, SAB3701239, Sigma
Streptavidin/HRP, 554066, BD Biosciences
anti-mouse IgG, 115-035-062, Jackson ImmunoResearch
anti-rabbit IgG, 111-035-045, Jackson ImmunoResearch
Alexa594-conjugated goat anti-rabbit IgG , A-11012, Thermo Fisher
Alexa594-conjugated rabbit anti-goat IgG , A-11037, Thermo Fisher
Alexa647-conjugated goat anti-rat IgG , A-21247, Thermo Fisher
Alexa488-conjugated goat anti-mouse IgG, A-28175, Thermo Fisher
anti-NeuN Antibody,Alexa Fluor®488 conjugated, MAB377X, Merck

**Validation**

Validation information of commercial antibodies is extracted from manufacturer's websites and listed in the format (abbrevations are explained at the bottom of this field):
Target/Name, 1°/2°/conjugated 1°+2° AB, Host, Reactivity, Applications-Manufacturer, Application-Manuscript, Catalog # , Clone (if applicable), Dilution used in the manuscript, Company, Citation PMID, Application in paper different from manufacturer's suggestion (Yes/No), Citation contains validation of application from paper (Yes/No)

anti-Fd, 1, S, H, Immunoelectrophoresis (IEP), Radial Immunodiffusion (RID), ELISA, PC075, n/a, 1:1000, Binding Site, 32776637, Y, Y
anti-Myc-tag , 1, Rb, MycTag, IHC-Fr, IP, WB, IHC-P, ICC, Electron Microscopy, WB, AB9106, n/a, 1:500, Abcam, 33534797, N, Y
anti-monomeric NeonGreen, 1, M, mNeonGreen fluorescent protein derived from Branchiostoma lanceolatum, IF, ELISA, WB, 32F6-100, 32F6, 1:1000, Chromotek, 32014414, Y, Y
anti-phospho-eIF2α, 1, Rb, H M R Mk Dm, WB, IP, IHC, WB, 3398, D9G8, 1:1000, Cell Signaling Technologies, 34953853, N, Y
anti-eIF2α, 1, Rb, H M R Mk, WB, IP, IHC, WB, 5324, D7D3, 1:1000, Cell Signaling Technologies, 34953853, N, Y

anti-Actin Antibody, 1, M, All, ELISA, ICC, IHC, IH(P), WB, WB, MAB1501R, C4, 1:10000, Millipore, 25753659, N, Y
anti-GFAP, 1, Rb, All, IHC, IHC, Z0334, n/a, 1:500, DAKO/Agilent, 11379820, N, Y
anti-F4/80, 1, R, M, F IHC IF IP WB , IHC, MCA497G, Cl:A3-1, 1 µg/mL, Serotec/BioRad, 31189648, N, Y
anti-human F(ab')2-alkaline phosphatase conjugated antibody, 2, G, H, IHC, ELISA, WB, ELISA, SAB3701239, n/a, 1:5000, Sigma, 32776637, N, Y
Streptavidin/HRP, 2, Streptomyces avidinii, Biotin, ELISA, ELISA, 554066, n/a, 1:1000, BD Biosciences, 20506300, N, Y
anti-mouse IgG, 2, Goat, M, ELISA, ICC, IHC, WB, WB, 115-035-062, n/a, 1:10000, Jackson ImmunoResearch, 31640849, N, Y
anti-rabbit IgG, 2, Goat, R, ELISA, ICC, IHC, WB, WB, 111-035-045, n/a, 1:10000, Jackson ImmunoResearch, 30143626, N, Y
Alexa594-conjugated goat anti-rabbit IgG , 2, G, R, IHC, IHC, A-11012, n/a, 1:1000, Thermo Fisher, 35190564, N, Y
Alexa594-conjugated rabbit anti-goat IgG , 2, Rb, G, IHC, IHC, A-11037, n/a, 1:1000, Thermo Fisher, 35600918, N, Y
Alexa647-conjugated goat anti-rat IgG , 2, G, R, IHC, IHC, A-21247, n/a, 1:1000, Thermo Fisher, 34211179, N, Y
Alexa488-conjugated goat anti-mouse IgG, 2, G, M, IHC, IHC, A-28175, n/a, 1:250, 1.6 µg/mL, Thermo Fisher, 34855620, N, Y
anti-NeuN Antibody,Alexa Fluor®488 conjugated, 1+2, M, H M R, IHC, IHC, MAB377X, A60, 1:1000, Merck, 32776637, N, Y

Species abbreviations:
H-Human
M-Mouse
R-Rat
Mk-Monkey
Dm-D. melanogaster
Rb-Rabbit
All-All Species Expected

Applications abbreviations:
ELISA-Enzyme-linked immunosorbent assay
WB-Western Blot
IP-Immunoprecipitation
IHC-Immunohistochemistry
IF-Immunofluorescence
F-Flow Cytometr
ICC-Immunocytochemistry
IEP-Immunoelectrophoresis
RID-Radial Immunodiffusion

# Eukaryotic cell lines

Policy information about cell lines and Sex and Gender in Research

| | |
|---|---|
| Cell line source(s) | CAD5 cells were derived from Cath.a-differentiated (CAD) cells (Qi Y et al., J Neurosci 98), were established by Mahal et al., PNAS 2007 and were a kind gift from Charles Weissmann. Generation of CAD5 Prnp knock-out cells was described in Bardelli et al., PLOS Pathogens 2018. |
| Authentication | CAD5 cells were not authenticated after reception from Charles Weissmann. |
| Mycoplasma contamination | Cell lines were tested negative for mycoplasma contamination. |
| Commonly misidentified lines (See ICLAC register) | No commonly misidentified cell lines were used in the study |

# Animals and other research organisms

Policy information about studies involving animals; ARRIVE guidelines recommended for reporting animal research, and Sex and Gender in Research

| | |
|---|---|
| Laboratory animals | Please specify the age for all strains of mice used, here in the reporting summary.<br><br>Please provide information on housing conditions for the mice, describing dark/light cycle, ambient temperature and humidity in the manuscript.<br>We used the following animals (both male and female) for slice culture and in vivo toxicity assessment: C57BL/6J, Tga20 (described in Fischer et al., EMBO J 1996). Additionally, Prnp0/0 (ZH1, described in Büeler et al., Cell 1993) and Prnp0/0 mice (ZH3, described in Nuvolone et al.,  J Exp Med 2016) were used for slice culture experiments.  Mice were bred in high hygienic grade facilities and housed in groups of 3–5, under a 12 h light/12 h dark cycle (from 7 am to 7 pm) at 21±1°C, with sterilized food (Kliba No. 3431, Provimi Kliba, Kaiseraugst, Switzerland) and water ad libitum.<br><br>Animals were not selected for gender. Animals ages were as follows:<br>Slice culture: 9-12 days<br>MEMRI: 4 months<br>AAV and prion injection: 3 months |
| Wild animals | This study did not involve wild animals |
| Reporting on sex | We did not adjust for mouse gender because of lack of group-wise comparisons. Specifically, for Memri experiments, treatment and |

| Reporting on sex | control were injected into the same animals. For AAV and prion inoculations, we compared survival versus antibody expression from only one group. |
|---|---|
| Field-collected samples | This study did not involve field-collected samples |
| Ethics oversight | We conducted all animal experiments in strict accordance with the Swiss Animal Protection law and dispositions of the Swiss Federal Office of Food Safety and Animal Welfare (BLV). The Animal Welfare Committee of the Canton of Zurich approved all animal protocols and experiments performed in this study (animal permits 123, ZH90/2013, ZH120/16, ZH139/16). |

Note that full information on the approval of the study protocol must also be provided in the manuscript.

# Flow Cytometry

## Plots

Confirm that:

☒ The axis labels state the marker and fluorochrome used (e.g. CD4-FITC).

☒ The axis scales are clearly visible. Include numbers along axes only for bottom left plot of group (a 'group' is an analysis of identical markers).

☒ All plots are contour plots with outliers or pseudocolor plots.

☒ A numerical value for number of cells or percentage (with statistics) is provided.

## Methodology

| Sample preparation | CAD5 cells were cultured with 20mL Corning Basal Cell Culture Liquid Media-DMEM and Ham's F-12, 50/50 Mix supplemented with 10% FBS, Gibco MEM Non-Essential Amino Ac-ids Solution 1X, Gibco GlutaMAX Supplement 1X and 0.5mg/mL of Geneticin in T75 Flasks ThermoFisher at 37°C 5% CO2. 16 hours before treatment, cells were split into 96wells plates at 25000 cells/well in 100μL.<br>POM1 alone was prepared at 5 μM final concentration, in 20 mM HEPES pH 7.2 and 150 mM NaCl. 100 μL of each sample, including buffer control, were added to CAD5 cells, in duplicates.<br>After 48 hours, cells were washed two times with 100μL MACS buffer (PBS + 1% FBS + 2 mM EDTA) and resuspended in 100 μL MACS buffer. 30'' before FACS measurements PI (1 μg/mL) was added to cells |
|---|---|
| Instrument | BD LSRFortessa |
| Software | FlowJo (10) |
| Cell population abundance | Only CAD5 cells are present in the sample |
| Gating strategy | Based on FSC and SSC the CAD5 cells were selected and separated from debris. After the first gating the cells were analyzed for PI presence. The gate to discriminate PI positive and negative cells was selected on control samples (not treated) and applied to all the other samples. A figure exemplifying the gating strategy is provided in ED Figure 3A. |

☒ Tick this box to confirm that a figure exemplifying the gating strategy is provided in the Supplementary Information.

# Magnetic resonance imaging

## Experimental design

| Design type | No BOLD imaging / fMRI |
|---|---|
| Design specifications | No BOLD imaging / fMRI |
| Behavioral performance measures | No behavioural performance measures were aquired. |

## Acquisition

| Imaging type(s) | Diffusion weighted imaging |
|---|---|
| Field strength | 4.7 |
| Sequence & imaging parameters | TR: 300 ms TE: 28 ms, flip angle: 90 deg, average: 1, Matrix: 350 x 350, Field of View: 3 x 3 cm, acquisi-tion time: 17 min, voxel size: 87x87 μm3, slice thickness: 700 μm3, Isodistance: 1400 μm3 and b values: 13, 816 s/mm2 |
| Area of acquisition | Converging on the whole hippocampus |
| Diffusion MRI | ☒ Used    ☐ Not used |
| Parameters | Single shell, b values: 13, 816 s/mm2 |

## Preprocessing

| | |
|---|---|
| Preprocessing software | Biomap software |
| Normalization | *If data were normalized/standardized, describe the approach(es): specify linear or non-linear and define image types used for transformation OR indicate that data were not normalized and explain rationale for lack of normalization.* |
| Normalization template | n/a |
| Noise and artifact removal | n/a |
| Volume censoring | n/a |

## Statistical modeling & inference

| | |
|---|---|
| Model type and settings | no fMRI |
| Effect(s) tested | no fMRI |

Specify type of analysis: ☐ Whole brain ☐ ROI-based ☐ Both

| | |
|---|---|
| Statistic type for inference<br>(See Eklund et al. 2016) | no fMRI |
| Correction | no fMRI |

## Models & analysis

| n/a | Involved in the study |
|---|---|
| ☒ ☐ | Functional and/or effective connectivity |
| ☒ ☐ | Graph analysis |
| ☒ ☐ | Multivariate modeling or predictive analysis |

