## [Peer Review File · Nature Structural & Molecular Biology]

Peer Review Information

Journal: Nature Structural and Molecular Biology

Manuscript Title: A conformational switch controlling the toxicity of the prion protein

Corresponding author name(s): Dr Karl Frontzek, Dr Luca Varani , Professor Adriano Aguzzi

Reviewer Comments & Decisions:

Decision Letter, initial version:
--

8th Feb 2022

Dear Karl,

Thank you again for submitting your manuscript "A conformational switch controlling the toxicity of the prion protein". I apologize for the delay in responding, which resulted from the difficulty in obtaining suitable referee reports. Nevertheless, we now have comments (below) from the 2 reviewers who evaluated your paper. In light of those reports, we remain interested in your study and would like to see your response to the comments of the referees, in the form of a revised manuscript.

I hope you will be pleased to see that both reviewers are very supportive and only make minor suggestions for improving the manuscript. Please be sure to address/respond to all concerns of the referees in full in a point-by-point response and highlight all changes in the revised manuscript text file. Please also note the additional guidance on our article format below. If you have comments that are intended for editors only, please include those in a separate cover letter.

We expect to see your revised manuscript within 6 weeks. If you cannot send it within this time, please contact us to discuss an extension; we would still consider your revision, provided that no similar work has been accepted for publication at NSMB or published elsewhere.

We are committed to providing a fair and constructive peer-review process. Do not hesitate to contact us if there are specific requests from the reviewers that you believe are technically impossible or

unlikely to yield a meaningful outcome.

When revising the manuscript, here are some guidelines for our Article format that you should follow to accelerate the further processing of your paper:

- the Abstract should be under 150 words, with no references
- main text is typically between 3,000 and 4,000 words (max. 4,500 words), and should be organized as Introduction, Results (with at least one subheading, <60 characters) and Discussion.
- We can accommodate up to 8 display items (Figures or Tables) in the main article and up to 10 Extended Data figures, which will be integrated into the full-text HTML version of your paper and will be appended to the online PDF. Each Extended Data item must be cited in order in the main text. Each Figure, Table and Extended Data figure must fit easily within an A4 page (210 x 297 mm), ideally with legends. Please ensure that the number and size of your Figures, Tables and Extended Data figures fulfil these requirements.
- In accordance with the above, we suggest converting the 10 most essential Supplementary figures to Extended Data Figures, because they are more easily accessible to readers.
- Supplementary material (text, figures, tables) should be submitted as part a single PDF file. There is no strict limit for Supplementary material.
- Author contributions should be supplied as a short text, not as table.
- Please prepare images of uncropped blots, we will ask you to provide these as Source Data later (see below).

Reporting Summary:

Please note that all key data shown in the main figures as cropped gels or blots should be presented in uncropped form, with molecular weight markers. These data can be aggregated into a single supplementary figure item. While these data can be displayed in a relatively informal style, they must refer back to the relevant figures. These data should be submitted with the final revision, as source data, prior to acceptance, but you may want to start putting it together at this point.

Data availability: this journal strongly supports public availability of data. All data used in accepted papers should be available via a public data repository, or alternatively, as Supplementary Information. If data can only be shared on request, please explain why in your Data Availability Statement, and also in the correspondence with your editor. Please note that for some data types, deposition in a public repository is mandatory - more information on our data deposition policies and available repositories

can be found below:

<https://www.nature.com/nature-research/editorial-policies/reporting-standards#availability-of-data>

[REDACTED]

Kind regards,
Florian

Florian Ullrich, Ph.D.
Associate Editor
Nature Structural & Molecular Biology
ORCID 0000-0002-1153-2040

Referee expertise:

Referee #1: structural biology, prions

Referee #2: neurodegenerative diseases, prions

Reviewers' Comments:

Reviewer #1:

Remarks to the Author:

This manuscript describes studies to evaluate the proposed role of a conformational switch in the prion protein (PrP), involving formation of an R208-H140 hydrogen bond ("H-latch") in the prion protein-mediated toxicity. Of important note, it has been demonstrated through many previous studies that PrP participation is sine qua non for the pathogenic effect of prions. Thus, while the prion (PrPSc), a misfolded conformer of PrPC, is the pathogen, it must directly interact with PrPC to exert its toxicity. The Authors, following the steps of their previous studies, take advantage of the interaction of some toxic anti-PrP antibodies that act as mimics of PrPSc to elucidate the molecular determinants of the pathogenic PrPSc/PrPC interaction. Of course, elucidating the molecular mechanisms of the pathogenesis of an invariably fatal disease is of the utmost interest. More so if that disease is the archetype of a novel category of maladies: prion diseases. Finally, a number of studies have lent support (although far from definitive) to the notion that Alzheimer's disease also involves PrP as a mediator, precisely acting as a "receptor" of pathogenic/toxic amyloid beta misfolded conformers.

In this context, the Authors have utilized an array of classic experimental approaches: 1) use of antibodies as ligand mimics; 2) generation of constitutively active ("latch mimic") and constitutively inactive ("latch dead") target (PrP) molecules; 3) generation of "latch dead" antibody ligands. By applying these reagents to adequate cell and animal models, and using appropriate cellular, biochemical and biophysical readouts, the Authors present solid and convincing evidence demonstrating that: 1) generation of the purported R208-H140 "H-latch" either by the effect of a ligand, or by covalent modification of PrP, leads to a deep conformational change in PrP with ripples propagating far beyond the R208.H140 pocket, and likely involving not only its globular domain but also its flexible tail (but vide infra); 2) such change is sufficient to induce toxicity; 3) such change is necessary to induce toxicity, as

reagents or covalent changes that block or impede it, abrogate toxicity.

Therefore, in my opinion the statement proposed by the title of the manuscript "A conformational switch controlling the toxicity of the prion protein (exists)" (my parenthesis), is fully supported by the experimental data. I have some additional comments:

1) The NMR spectra in Fig 3A are too small, which renders them uninformative. Compare them with Figure S4, in which the HSQC spectrum of the "latch mimic" PrP2Cys is clearly displayed. The spectra in Fig 3A should have a similar size. If this were not possible as a consequence of space limitations, they should be moved to the supplementary material. Also, the fact that the structure presented for the FT portion of PrP does not derive from NMR (like that of the GD portion) but rather is a reasonable model, should be highlighted in the figure legend. Incidentally, the changes induced by binding of the antibody to PrP are very extensive. This is also the case for those seen with more detail in Fig. S4 for PrP2Cys); to what extent is the general architecture of PrP preserved? Is PrP2Cys a slightly modified but recognizable version of wt PrP, or are whole portions of it structurally different? Of course I am not suggesting that the structure of PrP2Cys be solved, that would be outside the scope of the paper, although I think it would be a very interesting piece of information worthy of future studies. It is, though, relevant to the Authors' interpretation and discussion of the data. Thus, the Authors state (page 6, line 8) "This (the CD results) suggests that POM1 can alter the secondary structure of the FT". I disagree with this interpretation: as seen in Fig. 3B, the main difference between the calculated and CD-measured secondary structure components of the POM1-wtPrP23-231 ensemble is a decrease of alpha helix content from 51.7% to 31.5%, with a concomitant increase of the percentage of turns and "other" (coil) structure. Given that the FT is believed to contain mostly turn/coil secondary structure (not alpha helices), as adequately depicted in Fig 3A, the only possible explanation of this effect would be that a substantial portion of PrP23-231 alpha helices, about half of them, unfold to coil/turns upon binding to POM1. This assumes that most of the effect happens in PrP, not POM1. The fact that these changes do not take place if the FT is removed means that the FT is necessary to transmit the structural change initiated by the H-latch to the rest of the GD, but in my opinion cannot be interpreted as evidence of changes in the secondary structure of the FT.

2) The Authors aptly start their manuscript with a recapitulation of what is known (in part through their own contributions) about the chain of events linking PrPSc to its pathology: "The neurotoxicity of prions requires the interaction of the misfolded prion protein PrPSc with its cellular counterpart PrPC, which ultimately leading (should read "leads") to depletion of the PIKfyve kinase (1) and to spongiform encephalopathy". I believe it would be very adequate to close the circle, at the end of the manuscript, by discussing the findings in the context of this first statement. In other words, how do the Authors hypothesize that the H-latch-induced conformational change in PrP causes depletion of the PIKfyve kinase? In the cited paper (Lakkaraju et al.) a scheme (Fig. 7) links PrPSc infection with chronic ER stress leading to phosphorylation of eIF2alpha...which sets in motion a cascade of events eventually resulting

in decreased acetylation of PIKfyve. Therefore it seems that the conformational change induced by the H-latch in PrP should result in ER stress; how? Incidentally, the "latch mimic" PrP did not induce UPR stress when expressed in model cells.

3) Are the Author affiliations correct? To the best of my knowledge (but of course I might be missing a recent move) Tuomas Knowles is not affiliated with i3S-Universidade do Porto, but rather, to Cambridge University.

Reviewer #2:

Remarks to the Author:

It is well understood that infections prions trigger the conformational rearrangement of the normal cellular isoform of the prion protein, PrPC. The authors have previously identified an anti-PrPC monoclonal antibody (POM1) that, when bound, can trigger toxicity seemingly recapitulating the interaction of infectious prions with PrPC. In the current study, they characterize the conformational changes induced by POM1, which includes (but is not limited to) the formation of an intramolecular side-chain hydrogen bond R208-H140, dubbed an "H-latch," which alters the flexibility of two loops between $\alpha 2$ - $\alpha 3$ and $\beta 2$ - $\alpha 2$. Expression of a PrP2Cys mutant capable of covalent disulphide oxidation and mimicking the H-latch was constitutionally toxic, whereas a PrPR207A mutant unable to form the H-latch conferred resistance to prion infection. High-affinity ligands that prevented H-latch induction repressed prion-related neurodegeneration in organotypic cerebellar cultures. The authors selected recombinant "pomologs" scFv's that could bind wild-type PrPC, but not PrP2Cys. These binders depopulated H-latched conformers and conferred protection against prion toxicity. Finally, brain-specific expression of an antibody designed to prevent H-latch formation was significantly therapeutic in prion-infected mice (despite unhampered prion propagation).

This reviewer believes the manuscript is highly original, and with vast consequences for our scientific understanding of prion disease(s), and with significant translational applications. In addition to the new agents that can ameliorate prion toxicity, it should be noted that the authors have developed a strong rationale for a safety concern with the clinical anti-prion antibody ICSM18. The quality of the methodology and data seems impeccable across a large scope of computational and experimental approaches. The density of the manuscript is high, but the writing is clear without a wasted word. Statistics are appropriate according to this reviewer's understanding. The conclusions seem robust, and are in keeping with the statistical reproducibility of the data from multiple approaches.

A few typos were recognized by this reviewer:

page 2 line 15 "leads" instead of "leading"

page 7 lines 6,7 "Conversely, POM1 mutants retaining its affinity and epitope specificity but abolishing

H-latch formation" is not a sentence

page 8, lines 13,14 "hcY104A halted progression of prion toxicity even when they were already 13 conspicuous..." is not grammatical, because toxicity is singular, not plural

Is there a typo in panel B of Fig 2 on page 5? The + shows low lesion volume, and the 0 shows high lesion volume.

As to improvements, this reviewer has a few suggestions for consideration.

1. The name "H-latch" implies that the single low-energy hydrogen bond between R208-H140 is responsible for stabilization of the toxic conformation. The authors may wish to indicate that the H-latch is "reporting" on the conformational change, not its cause.
2. It is striking that prion toxicity and propagation can be dissociated -- can the authors speculate on how this occurs?
3. A few sentences on how the POM1-misfolded PrP^C is able to induce loss of PIKfyve kinase would be appropriate.

Author Rebuttal to Initial comments

Reviewers' Comments:

Reviewer #1:

Remarks to the Author:

This manuscript describes studies to evaluate the proposed role of a conformational switch in the prion protein (PrP), involving formation of an R208-H140 hydrogen bond ("H-latch") in the prion protein-mediated toxicity. Of important note, it has been demonstrated through many previous studies that PrP participation is sine qua non for the pathogenic effect of prions. Thus, while the prion (PrP^{Sc}), a misfolded conformer of PrP^C, is the pathogen, it must directly interact with PrP^C to exert its toxicity. The Authors, following the steps of their previous studies, take advantage of the interaction of some toxic anti-PrP antibodies that act as mimics of PrP^{Sc} to elucidate the molecular determinants of the pathogenic PrP^{Sc}/PrP^C interaction. Of course, elucidating the molecular mechanisms of the pathogeny of an invariably fatal disease is of the utmost interest. More so if that disease is the archetype of a novel category of maladies: prion diseases. Finally, a number of studies have lent support (although far from definitive) to the notion that Alzheimer's disease also involve PrP as a mediator, precisely acting as a "receptor" of pathogenic/toxic amyloid beta misfolded conformers.

In this context, the Authors have utilized an array of classic experimental approaches: 1) use of antibodies as ligand mimics; 2) generation of constitutively active ("latch mimic") and constitutively inactive ("latch dead") target (PrP) molecules; 3) generation of "latch dead" antibody ligands. By applying these reagents to adequate cell and animal models, and using appropriate cellular, biochemical and biophysical readouts, the Authors present solid and convincing evidence demonstrating that: 1) generation of the purported R208-H140 "H-latch" either by the effect of a ligand, or by covalent modification of PrP, leads to a deep conformational change in PrP with ripples propagating far beyond the R208.H140 pocket, and likely involving not only its globular domain but also its flexible tail (but vide infra); 2) such change is sufficient to induce toxicity; 3) such change is necessary to induce toxicity, as reagents or covalent changes that block or impede it, abrogate toxicity.

Therefore, in my opinion the statement proposed by the title of the manuscript "A conformational switch controlling the toxicity of the prion protein (exists)" (my parenthesis), is fully supported by the experimental data. I have some additional comments:

1) The NMR spectra in Fig 3A are too small, which renders them uninformative. Compare them with Figure S4, in which the HSQC spectrum of the "latch mimic" PrP^{2Cys} is clearly displayed. The spectra in Fig 3A should have a similar size. If this were not possible as a consequence of space limitations, they should be moved to the supplementary material.

Response: We have re-arranged the NMR spectra to be represented in a larger and readable format in addition to rearrangement of the figures according to editorial guidelines of Nature Structural & Molecular Biology as follows:

Original version	Revised version
Fig 1	Fig 1
Fig 2A, B	Fig 3D, E
Fig 2C-F	Fig 4A-D
Fig 3A-B	Fig 5

Fig 3C-E	Fig 6G-J
Fig 4	Fig 7
Fig S1	Extended Data Fig 1
Fig S2A-C	Fig 2A-C
Fig S2D-E	Extended Data Fig 2A-B
Fig S3A	Extended Data Fig 2E
Fig S3B,D	Extended Data Fig 2C-D
Fig S3C,E,F	Figure 2D-F
Fig S4	Fig 3A-C
Fig S5	Extended Data Fig 3A-D
Fig S6A	Extended Data Fig 3E
Fig S6B	Fig 3F
Fig S6C	Extended Data Fig 6B
Fig S6D-F	Extended Data Fig 6G-I
Fig S7	Extended Data Fig 4
Fig S8	Extended Data Fig 5A-B
Fig S9	Extended Data Fig 5C-D
Fig S10 (micrographs)	Supplementary Fig 1A
Fig S10 (bar graph)	Extended Data Fig 6A
Fig S11A	Supplementary Fig 1B
Fig S11B-C	Extended Data Fig 6C-D
Fig S12A-B	Extended Data Fig 6E-F
Fig S12C-E	Fig 4E-G
Fig S13	Supplementary Fig 2
Fig S14	Fig 6A-F
Fig S15	Extended Data Fig 7
Fig S16	Extended Data Fig 8
Fig S17	Extended Data Fig 9A
Fig S18	Extended Data Fig 9B
Fig S19	Extended Data Fig 10A
Fig S10	Extended Data Fig 10B

Also, the fact that the structure presented for the FT portion of PrP does not derive from NMR (like that of the GD portion) but rather is a reasonable model, should be highlighted in the figure legend.

Response: Thank you for noticing the shortcoming. The legend now reads:

“GD and part for the FT are shown on a molecular dynamics model of PrP” (Figure 5 in the revised version)

Incidentally, the changes induced by binding of the antibody to PrP are very extensive. This is also the case for those seen with more detail in Fig. S4 for PrP^{2Cys}); to what extent is the general architecture of PrP preserved? Is PrP^{2Cys} a slightly modified but recognizable version of wt PrP, or are whole portions of it structurally different? Of course I am not suggesting that the structure of PrP^{2Cys} be solved, that would be outside the scope of the paper, although I think it would be a very interesting piece of information worthy of future studies. It is, though, relevant to the Authors’ interpretation and discussion of the data.

Response: We too were intrigued by the extent of the NMR changes, especially since the PrP fold and secondary structure is virtually identical in the x-ray structure of PrP free and in complex with POM1. However, it is worth remarking that sidechain contacts between free and bound PrP are different also in the x-ray structure (see figure below, where contacts present only in free PrP are indicated by blue lines and those only in the PrP-POM1 complex in orange).

NMR chemical shifts are exquisitely sensitive to local and even transient conformations that escape x-ray analysis. We have extensive experience with this and have seen similar long range effects since our

early work in RNA-binding proteins (Varani L. et al.; Nat Struc Biol, 2000; <https://doi.org/10.1038/74101>), TCR-pMHC (Varani L. et al.; PNAS 2007; www.pnas.org/cgi/doi/10.1073/pnas.0703702104) and antibody-antigens (Wang J. et al.; Cell 2017; <https://doi.org/10.1016/j.cell.2017.09.002>; and more).

We believe that the NMR changes do not represent a change of fold but rather a shift in the populations of PrP conformers. This is supported by molecular dynamics simulations. The similarity between the long range NMR patterns in the POM1 complex and PrP^{2cys} supports the notion that they have similar structural properties, particularly in regards to the $\beta 2$ - $\alpha 2$ and $\alpha 2$ - $\alpha 3$ loop.

We confirmed that the overall structure remains the same with CD and immunohistochemical analysis probing the binding of several conformational antibodies to the POM1 complex and PrP^{2cys}. For instance, in line with NMR and MD, POM8 and POM19 recognize the $\alpha 1$ - $\alpha 2$ and $\beta 1$ - $\alpha 3$ regions in both free and bound PrP, whereas POM5 recognizes the $\beta 2$ - $\alpha 2$ loop only in free PrP. Furthermore, PrP^{2cys} shows glycosylation similar to PrP^{wt}. We conclude that PrP^{2cys} shares the overall fold of PrP^{wt} but has distinct, local conformational properties altering its flexibility and conformers populations.

Thank you for the suggestion to determine the structure of PrP^{2cys}, we are allocating the task to a student.

Thus, the Authors state (page 6, line 8) "This (the CD results) suggests that POM1 can alter the secondary structure of the FT". I disagree with this interpretation: as seen in Fig. 3B, the main difference between the calculated and CD-measured secondary structure components of the POM1-wtPrP₂₃₋₂₃₁ ensemble is a decrease of alpha helix content from 51.7% to 31.5%, with a concomitant increase of the percentage of turns and "other" (coil) structure. Given that the FT is believed to contain mostly turn/coil secondary structure (not alpha helices), as adequately depicted in Fig 3A, the only possible explanation of this effect would be that a substantial portion of PrP₂₃₋₂₃₁ alpha helices, about half of them, unfold to coil/turns upon binding to POM1. This assumes that most of the effect happens in PrP, not POM1. The fact that these changes do not take place if the FT is removed means that the FT is necessary to transmit the structural change initiated by the H-latch to the rest of the GD, but in my opinion cannot be

interpreted as evidence of changes in the secondary structure of the FT.

Response: Thank you for the profound observation. We had indeed thought about the same process. The FT might be acting in a way reminiscent of molten globules, with superposition of independent components in the CD signal. Simplifying, we can here refer to FT-changes or FT-dependent GD changes. We only mentioned the simpler and more direct hypothesis (FT-changes) in the paper. Indeed, some molecular dynamics simulations show the presence of transient helices in the FT, becoming less abundant in the complex. Other simulations, however, show transient contacts between the FT and GD, which would be in line with the hypothesis of FT-dependent GD changes. It must be stressed that although the FT-changes are more abundant in MD simulations, none of the above is statistically significant and sufficiently solid to be presented in a paper. It is also worth noting that the binding of conformational antibodies to the GD is not affected by POM1 binding, so it is not too easy to think about extensive alteration of the helices in the GD.

Either way, we believe that we are observing a shift in the populations of FT conformers with more toxic mutants/complexes having a higher percentage of toxic conformers.

In conclusion, we agree with the Reviewer's considerations and are now presenting both possibilities in the paper, which now reads on page 9, lines 20-29:

Circular-dichroism (CD) spectroscopy showed that the full rmPrP (rmPrP₂₃₋₂₃₁)-POM1 complex had more irregular structure content than its free components (Fig. 5B), whereas no difference was observed when POM1 was complexed to partially FT-deficient rmPrP₉₀₋₂₃₁. We did not observe any changes in the secondary structure of the ^{hc}Y104A-bound rmPrP₂₃₋₂₃₁ complex. This suggests that POM1 can alter the FT conformation with two possible mechanisms. Either i) the secondary structure of the FT itself is changed, probably through a shift in the population of conformers (FT-changes); ii) the secondary structure of the GD is altered in a FT-dependent manner, with FT-GD interactions stimulated by POM1 binding. Hence H-latch induction leads to subtle alterations of the structure of both GD and FT, whose presence correlates with toxicity.

2) The Authors aptly start their manuscript with a recapitulation of what is known (in part through their own contributions) about the chain of events linking PrP^{Sc} to its pathology: "The neurotoxicity of prions requires the interaction of the misfolded prion protein PrP^{Sc} with its cellular counterpart PrP^C, which ultimately leading (should read "leads") to depletion of the PIKfyve kinase (1) and to spongiform encephalopathy". I believe it would be very adequate to close the circle, at the end of the manuscript, by discussing the findings in the context of this first statement. In other words, how do the Authors hypothesize that the H-latch-induced conformational change in PrP causes depletion of the PIKfyve kinase? In the cited paper (Lakkaraju et al.) a scheme (Fig. 7) links PrP^{Sc} infection with chronic ER stress leading to phosphorylation of eIF2alpha...which sets in motion a cascade of events eventually resulting in decreased acetylation of PIKfyve. Therefore it seems that the conformational change induced by the H-latch in PrP should result in ER stress; how? Incidentally, the "latch mimic" PrP did not induce UPR stress when expressed in model cells.

Response: Bona fide prion infections as well as prion mimetics, e.g. toxic anti-PrP^C antibodies, were shown to induce the unfolded protein response (UPR), although the exact mechanisms are still at large (Moreno et al., 2012; Hermann et al., 2015). We have shown that prolonged UPR leads to PIKfyve deacylation and rapid depletion, resulting in endolysosomal hypertrophy (Lakkaraju et al., 2021). PIKfyve depletion was observed after prion infection and toxic anti-PrP^C antibodies in mice and organotypic cultured slices. However, treatment of neuronal monocultures with toxic anti-PrP^C antibodies did not induce ER stress or PIKfyve depletion (unpublished data), possibly because of concentrations of the antibodies used or lack of a prolonged exposure to the antibodies unlike prions themselves which are constantly being replicated on the cells. ER stress and PIKfyve depletion upon POM1 treatment is observed in COCS suggesting a role for the non-neuronal cell types or other cofactors in manifesting this phenotype. Gt1 cells exposed to prions show ER stress and PIKfyve depletion at 60 days post infection suggesting a long term exposure to prions is needed. It is highly probable that upon elevated and prolonged expression of PrP^{2Cys} in the cells also leads to a similar phenotype, however a detailed time course experiment will need to be designed to investigate this. It is also possible that these cells do not exert ER stress and could trigger the toxicity cascade in the cells bypassing the PIKfyve route. A further

detailed study on the mode of toxicity between PrP^{2Cys} and prions is warranted to address this question, which currently is beyond the scope of this manuscript.

We have amended the discussion (page 14 lines 15-21) to read as follows:

“Spongiform change, e.g. endolysosomal hypertrophy through UPR activation and subsequent PIKfyve depletion, is shared in both prion and POM1 toxicity (Lakkaraju et al.). Multiple toxic cascades are activated in prion infections and in cells treated with POM1 (Hermann et al.). Cells that stably express PrP^{2Cys} are not affected by UPR in the current experimental paradigm, suggesting that either the protein dosage is insufficient to observe UPR or its toxicity is independent of PIKfyve depletion. Besides neuronal loss, which is shared amongst prion, POM1 and PrP^{2Cys} toxicity, it will be interesting to investigate the overlap of toxic cascades between the different prion disease models, which could provide important knowledge of early disease-associated changes.”

3) Are the Author affiliations correct? To the best of my knowledge (but of course I might be missing a recent move) Tuomas Knowles is not affiliated with i3S-Universidade do Porto, but rather, to Cambridge University.

Response: We apologize for the error and have now assigned the correct affiliations.

Reviewer #2:

Remarks to the Author:

It is well understood that infections prions trigger the conformational rearrangement of the normal cellular isoform of the prion protein, PrP^C. The authors have previously identified an anti-PrP^C monoclonal antibody (POM1) that, when bound, can trigger toxicity seemingly recapitulating the interaction of infectious prions with PrP^C. In the current study, they characterize the conformational changes induced by POM1, which includes (but is not limited to) the formation of an intramolecular side-chain hydrogen bond R208-H140, dubbed an “H-latch,” which alters the flexibility of two loops between α 2- α 3 and β 2- α 2. Expression of a PrP^{2Cys} mutant capable of covalent disulphide oxidation and mimicking the H-latch was constitutionally toxic, whereas a PrP^{R207A} mutant unable to form the H-latch conferred resistance to prion infection. High-affinity ligands that prevented H-latch induction repressed prion-related neurodegeneration in organotypic cerebellar cultures. The authors

selected recombinant "pomologs" scFv's that could bind wild-type PrP^C, but not PrP^{2Cys}. These binders depopulated H-latched conformers and conferred protection against prion toxicity. Finally, brain-specific expression of an antibody designed to prevent H-latch formation was significantly therapeutic in prion-infected mice (despite unhampered prion propagation).

This reviewer believes the manuscript is highly original, and with vast consequences for our scientific understanding of prion disease(s), and with significant translational applications. In addition to the new agents that can ameliorate prion toxicity, it should be noted that the authors have developed a strong rationale for a safety concern with the clinical anti-prion antibody ICSM18. The quality of the methodology and data seems impeccable across a large scope of computational and experimental approaches. The density of the manuscript is high, but the writing is clear without a wasted word. Statistics are appropriate according to this reviewer's understanding. The conclusions seem robust, and are in keeping with the statistical reproducibility of the data from multiple approaches.

A few typos were recognized by this reviewer:

page 2 line 15 "leads" instead of "leading"

Response: This typo was corrected

page 7 lines 6,7 "Conversely, POM1 mutants retaining its affinity and epitope specificity but abolishing H-latch formation" is not a sentence

Response: This sentence now reads: "Conversely, POM1 mutants retaining its affinity and epitope specificity but abolishing H-latch formation proved neuroprotective." – page 14, lines 5+6

page 8, lines 13,14 "hcY104A halted progression of prion toxicity even when they were already 13 conspicuous..." is not grammatical, because toxicity is singular, not plural

Response: This sentence was corrected to singular.

Is there a typo in panel B of Fig 2 on page 5? The + shows low lesion volume, and the 0 shows high lesion

volume.

Response: We apologize for the unintuitive labeling. +/0 refers to pre-incubation with cognate antigen, e.g., rmPrP₂₃₋₂₃₁, as negative control, highlighting lesion volume is due to antibody injection but not when antibody is in complex with the antigen. We have now amended the legend which now reads “preincubation with rmPrP₂₃₋₂₃₁” (revised Figure 6G).

As to improvements, this reviewer has a few suggestions for consideration.

1. The name "H-latch" implies that the single low-energy hydrogen bond between R208-H140 is responsible for stabilization of the toxic conformation. The authors may wish to indicate that the H-latch is "reporting" on the conformational change, not its cause.

Response: We agree with the Reviewer that the term H-Latch directs towards the R208 H-bond, which is only one of the aspects of the toxic conformers that are also characterized by, for instance, altered flexibility in the $\beta 2-\alpha 2$ and $\alpha 2-\alpha 3$ loops. At the same time, we did not find a different term that could be easily and effectively utilized in a short manuscript.

We have altered the text to reflect the above. For instance, the last sentence of the abstract now reads “[...] confirming that the H-latch is an important reporter of prion neurotoxicity” instead of “[...] confirming that the H-latch is causally linked to prion neurotoxicity”.

Accordingly, the first paragraph of the discussion, page 14 lines 2+3 was changed to read *“In summary, the evidence presented here suggests that H-latch formation is an important feature of prion toxicity.”* instead of *“In summary, the evidence presented here suggests that H-latch formation is an important driver of prion toxicity.”*

2. It is striking that prion toxicity and propagation can be dissociated -- can the authors speculate on how this occurs?

Response: We consider ^{hc}Y104A to exert a dominant-negative effect on PrP^C: neurotoxicity of harmful pomologs can be halted by pre-engagement of antibodies against the flexible tail of PrP^C (PrP^C-FT) and neuroprotection of ^{hc}Y104A in prion-infected mice occurs despite of elevated PrP^{Sc} levels. This suggests that pomologs act downstream of PrP^{Sc}-PrP^C interaction but upstream of PrP^C-FT engagement. Due to pathological similarities of prions and prion-mimetic antibodies, one may consider the α 1- α 3 helix of PrP^C, e.g. the POM1 epitope, a binding site of PrP^{Sc}. We speculate that blockage of PrP^{Sc}-PrP^C interaction by ^{hc}Y104A caused unlinking of prion propagation and toxicity. We have amended the discussion on page 14, lines 31-34 by adding “Finally, intracerebrally injected ^{hc}Y104A was innocuous, and AAV-transduced ^{hc}Y104A extended the life span of prion-infected mice, despite elevated PrP^{Sc} levels, suggesting it acts downstream of PrP^{Sc} replication, possibly by blocking a PrP^{Sc}-PrP^C interaction at the POM1 epitope.”

3. A few sentences on how the POM1-misfolded PrP^C is able to induce loss of PIKfyve kinase would be appropriate.

Response: We have responded to a similar comment above, e.g. the second issue raised by reviewer 1: Bona fide prion infections as well as prion mimetics, e.g. toxic anti-PrP^C antibodies, were shown to induce the unfolded protein response (UPR), although the exact mechanisms are still at large (Moreno et al., 2012; Hermann et al., 2015). We have shown that prolonged UPR leads to PIKfyve deacylation and rapid depletion, resulting in endolysosomal hypertrophy (Lakkaraju et al., 2021). PIKfyve depletion was observed after prion infection and toxic anti-PrP^C antibodies in mice and organotypic cultured slices. However, treatment of neuronal monocultures with toxic anti-PrP^C antibodies did not induce ER stress or PIKfyve depletion (unpublished data), possibly because of concentrations of the antibodies used or lack of a prolonged exposure to the antibodies unlike prions themselves which are constantly being replicated on the cells. ER stress and PIKfyve depletion upon POM1 treatment is observed in COCS suggesting a role for the non-neuronal cell types or other cofactors in manifesting this phenotype. Gt1 cells exposed to prions show ER stress and PIKfyve depletion at 60 days post infection suggesting a long term exposure to prions is needed. It is highly probable that upon elevated and prolonged expression of

PrP^{2Cys} in the cells also leads to a similar phenotype, however a detailed time course experiment will need to be designed to investigate this. It is also possible that these cells do not exert ER stress and could trigger the toxicity cascade in the cells bypassing the PIKfyve route. A further detailed study on the mode of toxicity between PrP^{2Cys} and prions is warranted to address this question, which currently is beyond the scope of this manuscript.

We have amended the discussion (page 14 lines 15-21) to read as follows:

“Spongiform change, e.g. endolysosomal hypertrophy through UPR activation and subsequent PIKfyve depletion, is shared in both prion and POM1 toxicity (Lakkaraju et al.). Multiple toxic cascades are activated in prion infections and in cells treated with POM1 (Hermann et al.). Cells that stably express PrP^{2Cys} are not affected by UPR in the current experimental paradigm, suggesting that either the protein dosage is insufficient to observe UPR or its toxicity is independent of PIKfyve depletion. Besides neuronal loss, which is shared amongst prion, POM1 and PrP^{2Cys} toxicity, it will be interesting to investigate the overlap of toxic cascades between the different prion disease models, which could provide important knowledge of early disease-associated changes.”

Decision Letter, first revision:

Apr 20, 2022

Dear Karl,

Thank you for submitting your revised manuscript "A conformational switch controlling the toxicity of the prion protein" (NSMB-A45716A). It has now been seen by one of the original referees who finds that the paper has improved in revision. Therefore we'll be happy in principle to publish it in Nature Structural & Molecular Biology, pending minor revisions to comply with our editorial and formatting guidelines.

Kind regards,

Florian

Florian Ullrich, Ph.D.
Associate Editor
Nature Structural & Molecular Biology
ORCID 0000-0002-1153-2040

Reviewer #1 (Remarks to the Author):

The Authors have thoroughly and convincingly responded to all my queries.

Final Decision Letter: